# Adaptation of a transmitted/founder simian-human immunodeficiency virus for enhanced replication in rhesus macaques

Anya Bauer[1], Emily Lindemuth[1], Francesco Elia Marino[1], Ryan Krause[1], Jaimy Joy[1],
Steffen S. Docken[2], Suvadip Mallick[1], Kevin McCormick[1], Clinton Holt[3], Ivelin Georgiev[3],
Barbara Felber[4], Brandon F. Keele[5], Ronald Veazey[6], Miles P. Davenport[2], Hui Li[1,7],
George M. Shaw[1,7], Katharine J. Bar[1] *

1 Department of Medicine, Perelman School of Medicine, University of Pennsylvania, Philadelphia,
Pennsylvania, United States of America, 2 Kirby Institute, University of New South Wales, Sydney, Australia,
3 Department of Pathology, Microbiology, and Immunology, Vanderbilt University, Nashville, Tennessee,
United States of America, 4 Human Retrovirus Pathogenesis Section, Vaccine Branch, Center for Cancer
Research, National Cancer Institute, Maryland, United States of America, 5 AIDS and Cancer Virus Program,
Frederick National Laboratory for Cancer Research, Frederick, Maryland, United States of America,
6 Department of Pathology and Laboratory Medicine, Tulane School of Medicine, New Orleans, Louisiana,
United States of America, 7 Departments of Microbiology, Perelman School of Medicine, University of
Pennsylvania, Philadelphia, Pennsylvania, United States of America

* bark@pennmedicine.upenn.edu

UNITED STATES

**Data Availability Statement:** All relevant data are
within the paper and its Supporting information
files.

## Abstract

Transmitted/founder (TF) simian-human immunodeficiency viruses (SHIVs) express HIV-1
envelopes modified at position 375 to efficiently infect rhesus macaques while preserving
authentic HIV-1 Env biology. SHIV.C.CH505 is an extensively characterized virus encoding
the TF HIV-1 Env CH505 mutated at position 375 shown to recapitulate key features of HIV-
1 immunobiology, including CCR5-tropism, a tier 2 neutralization profile, reproducible early
viral kinetics, and authentic immune responses. SHIV.C.CH505 is used frequently in nonhu-
man primate studies of HIV, but viral loads after months of infection are variable and typically
lower than those in people living with HIV. We hypothesized that additional mutations
besides Δ375 might further enhance virus fitness without compromising essential compo-
nents of CH505 Env biology. From sequence analysis of SHIV.C.CH505-infected macaques
across multiple experiments, we identified a signature of envelope mutations associated
with higher viremia. We then used short-term *in vivo* mutational selection and competition to
identify a minimally adapted SHIV.C.CH505 with just five amino acid changes that substan-
tially improve virus replication fitness in macaques. Next, we validated the performance of
the adapted SHIV *in vitro* and *in vivo* and identified the mechanistic contributions of selected
mutations. *In vitro*, the adapted SHIV shows improved virus entry, enhanced replication on
primary rhesus cells, and preserved neutralization profiles. *In vivo*, the minimally adapted
virus rapidly outcompetes the parental SHIV with an estimated growth advantage of 0.14
days$^{-1}$ and persists through suppressive antiretroviral therapy to rebound at treatment inter-
ruption. Here, we report the successful generation of a well-characterized, minimally
adapted virus, termed SHIV.C.CH505.v2, with enhanced replication fitness and preserved

**Funding:** The study was supported by the following grants: Office of AIDS Research P01-AI131338 to KJB, Office of AIDS Research R01-AI162646 to KJB, Beat HIV Collaboratory UM1AI164570 to KJB, Collaboratory of AIDS Researchers for Eradication UM1AI126619 to KJB. The funders had no role in study design, data collection, and analysis, the decision to publish, or preparation of the manuscript.

**Competing interests:** The authors have declared that no competing interests exist.

native Env properties that can serve as a new reagent for NHP studies of HIV-1 transmission, pathogenesis, and cure.

## Author summary

The power of the nonhuman primate model of HIV to predict outcomes in people living with HIV (PLWH) depends on authentic virus-host interactions. In pursuit of viruses that generate infection that mirrors the effects of HIV-1 in PLWH, we developed a minimally adapted version of a commonly used virus, SHIV.C.CH505, which has better fitness than the parental virus while retaining important biological properties. First, we studied virus sequences from SHIV.C.CH505-infected rhesus macaques to identify a signature of mutations common to animals with higher viral loads. We then tested viruses containing the various mutations in the lab and in animals to determine the most fit version and to identify the contribution of each mutation. Ultimately, we identified a minimally adapted version of SHIV.C.CH505 with just 5 amino acid substitutions that enhances virus replication and preserves CH505 envelope properties, including sensitivity to clinically relevant broadly neutralizing antibodies. This new virus, called SHIV.C.CH505.v2, replicates well in macaques over time and persists through antiretroviral therapy. SHIV.C.CH505.v2 could be an important component of nonhuman primate studies of HIV prevention, therapy, and cure.

## Introduction

Nonhuman primate (NHP) models of HIV-1 (human immunodeficiency virus-1) infection are integral to elucidating basic mechanisms of virus transmission, immunopathogenesis, prevention and cure [1–3]. SIVmac (simian immunodeficiency virus from macaque) infection of rhesus macaques (RM) consistently leads to high-level viremia and accelerated disease progression, making it a widely used model of HIV/AIDS [4]. However, differences between SIVmac and HIV-1, including highly divergent envelope (*env*) sequences, distinct accessory genes, and variable set-point viremia and natural history, challenge the direct translation of NHP study findings to people living with HIV-1 (PLWH) [5,6]. For studies of HIV-1 Env targeting agents, chimeric simian-human immunodeficiency viruses, or SHIVs, which encode HIV-1 Env within an SIV backbone of structural and regulatory elements, have distinct benefits. Thus, SHIVs have been used extensively to develop antibody-based vaccines, broadly neutralizing antibodies (bnAbs), and other Env-targeted approaches.

First developed decades ago, SHIVs have a complex legacy of both important discoveries and misleading findings [7]. Early SHIVs encoded HIV-1 Envs from lab-adapted viruses that were syncytiopathic and subsequently determined to be CXCR4 or dual CXCR4/CCR5 tropic [8–11]. Other SHIVs were derived from primary HIV-1 strains but replicated in RMs efficiently only after extensive *in vitro* or *in vivo* passage [12–17]. Rare primary HIV-1 Envs supported efficient SHIV replication in RMs without adaptation [18–20]. Thus, most early SHIVs differed from primary HIV-1 isolates in fundamental properties such as coreceptor-tropism, neutralization sensitivity, and pathogenicity [10]. Nonetheless, SHIVs have contributed prominently as a model of HIV-1 immunobiology and for vaccine and cure research [21–28].

In 2008, a novel experimental strategy was developed based on single genome sequencing of plasma viral RNA from acute infection samples, which allowed for a precise and unambiguous identification of viruses responsible clinical transmission of HIV-1 [29,30]. These and subsequent analyses of HIV-1 transmitted/founder (TF) env genes [29,30] and full-length viral genomes [31,32], showed these viruses to be uniformly CD4 dependent and CCR5-tropic, with tier 2 neutralization resistance and enhanced resistance to type I interferons [33–38]. The term TF virus was coined to distinguish such viruses from virus isolates or cloned viral genomes that were identified by other means or at other clinical timepoints. However, like primary HIV-1 Envs, TF Envs generally could not support SHIV replication in RMs. That changed in 2016 with the discovery that substitution of a single amino acid (residue 375, numbering according to HXB2 sequence) in the CD4-binding pocket of HIV-1 Env enabled efficient SHIV replication in rhesus cells [39]. Importantly, this mutation did not substantially alter the antigenicity, neutralization sensitivity, fusogenicity or cell tropism of the parental Env [39]. To determine the optimal Env375 amino acid for each TF SHIV, investigators used *in vivo* competition of SHIV variants encoding either the wildtype serine (or threonine) at position 375 or one of five distinct bulky, hydrophobic, or aromatic residues. With this novel strategy, a panel of 16 SHIVs was generated, each encoding a TF or primary HIV-1 Env, including some such as BG505 and CH505 that are of particular interest to the vaccine and cure fields [40].

TF SHIVs demonstrate key features of HIV-1 immunobiology and have increasingly been used as reagents in the NHP model. Like TF HIV-1 strains [29,33], TF SHIVs are CCR5-tropic and consistently confer productive infection after parenteral or mucosal inoculation. TF SHIV infections reliably reproduce features of acute HIV-1 infection, including high titer peak viremia at about 14 days post-inoculation, and settle to a setpoint of $10^3–10^4$ vRNA/ml several weeks later. TF SHIVs consistently elicit autologous neutralizing antibody (nAb) responses similar to HIV-1 infection, with molecular patterns of Env-antibody coevolution that mirror those of HIV-1 infection in humans, including the elicitation of broadly neutralizing antibodies (bnAbs) in some animals [41]. In a subset of TF SHIVs that have been tested, infection persists through suppressive antiretroviral therapy and rebounds to near plasma viral load setpoints upon treatment interruption [42]. These similarities between SHIV and HIV-1 infections notwithstanding, viral load setpoints in SHIV-infected RMs are variable, but typically lower than those seen in PLWH and markedly lower than SIVmac-infected RM, and the proportion of animals with undetectable SHIV viremia is higher.

SHIV.C.CH505 is an extensively characterized CCR5-tropic TF SHIV that possesses a tier 2 neutralization phenotype [29,40]. With consistent early viral replication kinetics and authentic adaptive immune responses, SHIV.C.CH505 has become a valued reagent in immunopathogenesis and vaccine challenge studies [7,43]. While peak and early viral kinetics mirror HIV-1, SHIV.C.CH505-infected RMs often display lower viral loads after months of infection and relatively frequent spontaneous control. We hypothesized that additional mutations besides Δ375 might further enhance virus fitness without compromising essential components of CH505 Env biology. We used short-term *in vivo* mutational selection and competition to identify a minimally adapted SHIV.C.CH505 with the fewest possible mutations required to improve virus replication fitness in macaques. Next, we validated the performance of this SHIV *in vitro* and *in vivo* and identified the mechanistic contributions of selected mutations. Here, we report the successful generation of a well-characterized, minimally adapted virus, termed SHIV.C.CH505.v2, with enhanced replication fitness and preserved native Env properties that can serve as a new reagent for NHP studies of HIV-1 transmission, pathogenesis, and cure.

## Results

### Identification of a sequence signature of enhanced replication in SHIV.C. CH505 infected RM

The determinants of viral replication and setpoint in HIV-1 infection, and similarly in SIV/ SHIV infection, are multifactorial, balancing intrinsic virus fitness and susceptibility to host immune pressures. TF viruses from an acutely infected individual can carry with them mutations acquired in the previous host, including escape mutations from adaptive immune responses [29,44,45]. Such mutations provide enhanced fitness in the previous host but not necessarily in the new recipient. This is the explanation for reversion to the global consensus [46–49], which is a well-described phenomenon in acute HIV-1 infection. For SHIVs, which bear an HIV-1 Env adapted for replication fitness in humans, the challenges for replication in RMs whose CD4 and CCR5 molecules differ substantially from humans, may be even more extreme. Like HIV-1 and many TF SHIVs, SHIV.C.CH505 infection across multiple animals results in a range of virus setpoints. Across more than 100 SHIV.C.CH505-infected adult RM in 7 published studies [39,43,50–54], approximately 20% of animals controlled virus to below $10^2$ copies/ml within the first 3–6 months of infection whereas approximately 25% of macaques maintained consistent viral loads over $10^4$ copies/ml. From the quartile of macaques experiencing higher viral loads (VL), we assessed virus evolution in three SHIV.C. CH505-infected RM exhibiting sustained viremia between $10^4$ to $10^6$ copies/ml for at least 6 months (Fig 1B). To minimize the assessment of virus adaptation to host-specific cellular immune responses (*eg*, HLA-restricted CTL epitope adaptation), we cataloged mutations that were shared across all three outbred RM. Single genome sequencing (SGS), which precludes *in vitro* recombination, enables linkage across amplicons, and allows proportional assessment of diverse virus populations [29,32], was used to generate sequences of gp160 *env* from plasma virus collected at weeks 4, 10, and 20 post-infection (total 287 sequences, median 41 per animal timepoint) (Fig 1A). As in early infection in PLWH and in acutely SIV/SHIV-infected RM, week 4 sequences largely reflect the TF virus (SHIV.C.CH505) with only sporadic, random mutations acquired throughout the genome [29,32,54]. By weeks 10 and 20 post-infection, however, shared pathways of virus adaptation [45] were apparent, with six shared mutations common across all animals. Two mutations, N334S and H417R, arose early, were present in sequences of all RM by week 10, and persisted through week 20. Four additional mutations (N130D, N279D, K302N, and Y330H) arose later and displayed high frequency in all three RM (>90% for Y330H and >50% for the others) (Fig 1C). Thus, six sites of shared selection pressure in gp160 *env* (130, 279, 302, 330, 334, and 417) were identified across the three high VL RM. Except for N334S, which arose early in the vast majority of SHIV.C.CH505-infected RM, these mutations were less frequently selected in RM with lower-level viremia (S1 Fig).

### Confirmation of signature mutations in a second cohort of SHIV.C.CH505 infected RM

We next evaluated longitudinal sequences from a distinct SHIV.C.CH505 experiment to confirm the identified mutational signature. We studied a vaccine challenge experiment, in which 45 RM were immunized with a CH505 Env DNA vaccine or control, and then mucosally challenged with SHIV.C.CH505 [50]. This vaccine strategy demonstrated neither protective efficacy nor durable alteration in setpoint viremia between study arms. At 52 weeks post-infection, SGS of gp160 *env* was performed in 22 SHIV.C.CH505-infected RM, of which five RM had average viremia over $10^4$ copies/ml at timepoints after study week 12. At

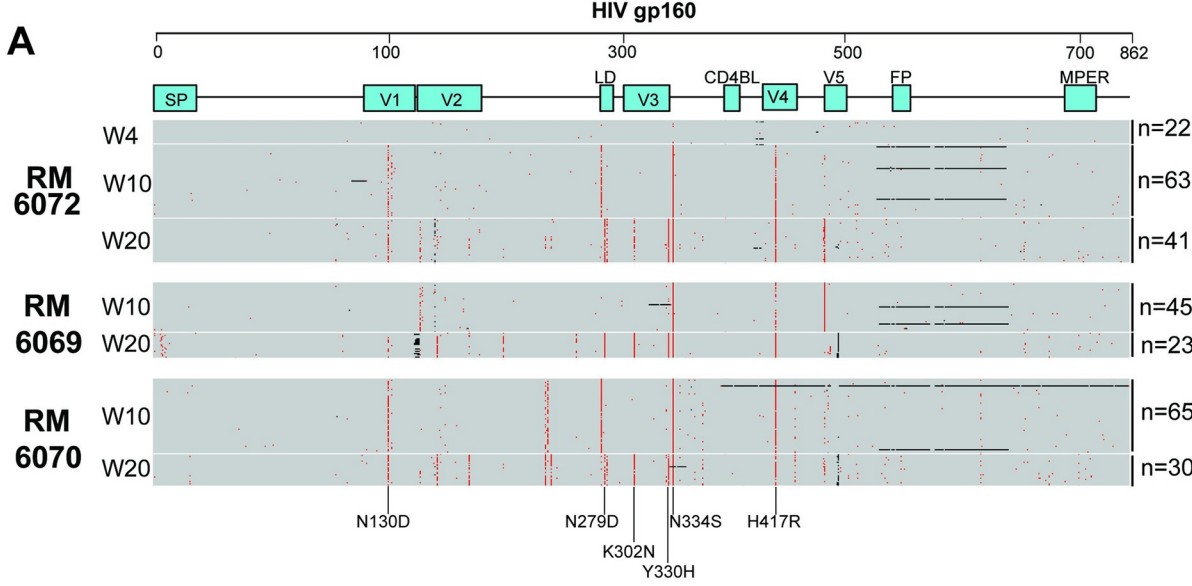

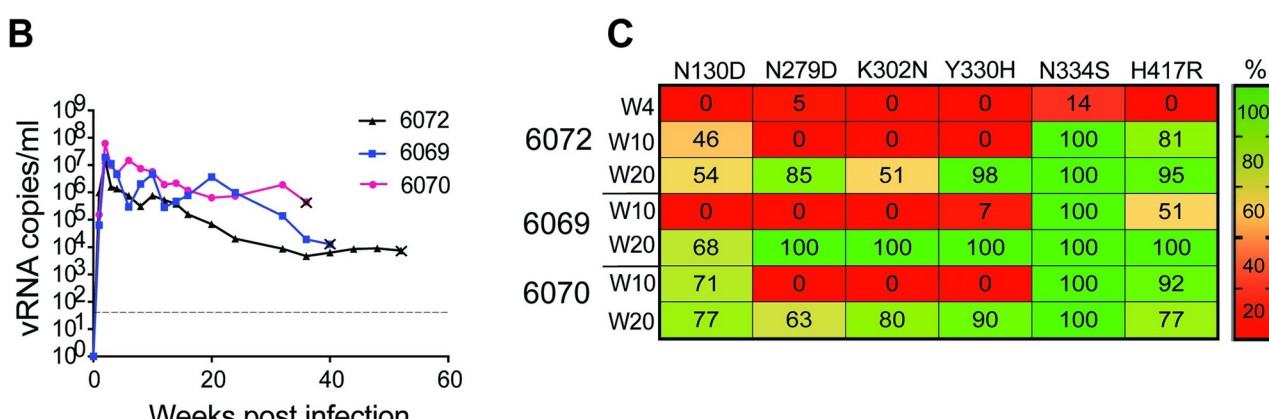

**Fig 1. Identification of a signature of shared mutations in animals with higher viral loads.** (**A**) Env sequences through 20 weeks of infection in three RM with higher viral load setpoints reveal several shared mutations. Amino acid highlighter plot of SGS-derived gp160 Env sequences generated from plasma virus collected at weeks 4, 10, and 20 post-infection in SHIV.C.CH505-infected RM, with SHIV.C.CH505.TF as master sequence. Red tics indicate amino acid substitutions, with six high frequency mutations seen across RM annotated below. Black tics indicate deletions. (**B**) Plasma viral load determined by quantitative real-time reverse-transcription PCR. The dotted line indicates the assay's limit of detection (LoD) (62 copies/ml). Euthanasia timepoints are marked with the x symbol. (**C**) Frequency of the six shared mutations through 20 weeks.

52 WPI, we assessed the frequency of RM in which mutations were fully penetrant (*i.e.*, 100% of sequences) in RM with high vs. low VL. The mutation N334S was frequently penetrant across RM (100% of higher VL RM vs. 80% of lower VL RM). Three signature mutations (K302N, Y330H, and H417R) were fully penetrant in four out of five high VL RM, and substantially less penetrant in the lower VL RM. The mutations N130D and N279D were fully penetrant in a lower proportion of animals in both groups (Fig 2 and S1 Fig). Thus, a similar Env mutational signature of higher viremia was identified in the second cohort of SHIV.C.CH505-infected RM.

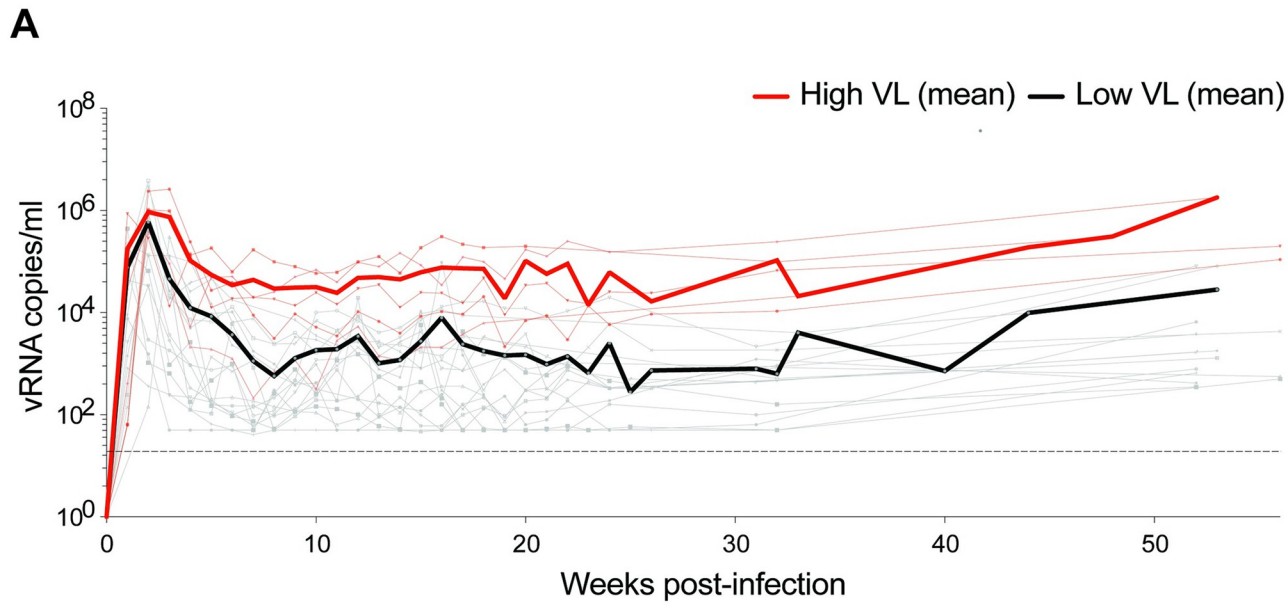

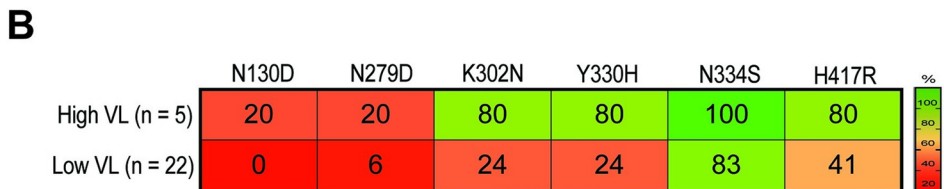

**Fig 2. Signature confirmation in a second SHIV.C.CH505-infected RM cohort.** (**A**) *In vivo* viral plasma kinetics from a published vaccination study of 22 RM immunized with a CH505 Env DNA vaccine or sham control and challenged with SHIV.C.CH505 [50]. Average viremia threshold of >$10^4$ copies/ml distinguished animals with higher viral loads (n = 5) from those with lower viral loads (n = 17). Geometric mean viral loads are indicated by the thick red and black lines, respectively. The dotted line indicates the assay's limit of detection (LoD) (62 copies/ml). (**B**) Percentage of RM with 100% penetrance of specified mutations determined by SGS of gp160 env in high vs. low VL RM.

## *In vivo* competition of wildtype virus and signature mutants

To assess the relative fitness of viruses containing the signature mutations against SHIV.C. CH505.TF virus, we performed two successive *in vivo* competition experiments. First, we intravenously inoculated RM 5695 with the early plasmas from the three SHIV.C.CH505-infected RM described above (200 µl of plasma from weeks 4, 10 and 20 post-infection from RM6069, 6070, 6072; 1.4 mL *in toto*; Fig 3A) [41]. RM 5695 experienced rapid, high peak viremia (>$10^6$ c/ml) and maintained high-level setpoint viremia for over a year, until it was euthanized for progression to simian AIDS (Fig 3A). A total of 194 SGS-derived gp160 *env* sequences (median 42 per timepoint) were derived over the first six months of infection to assess the proportion of signature mutations over time (Fig 3B). As expected with a high-titer IV administration of a diverse virus inoculum, week 2 sequencing revealed infection of multiple unique TF viruses [29,54], with proportions reflecting those of the inoculum. By week 4, however, there was a substantial shift in replicating virus populations, with those containing

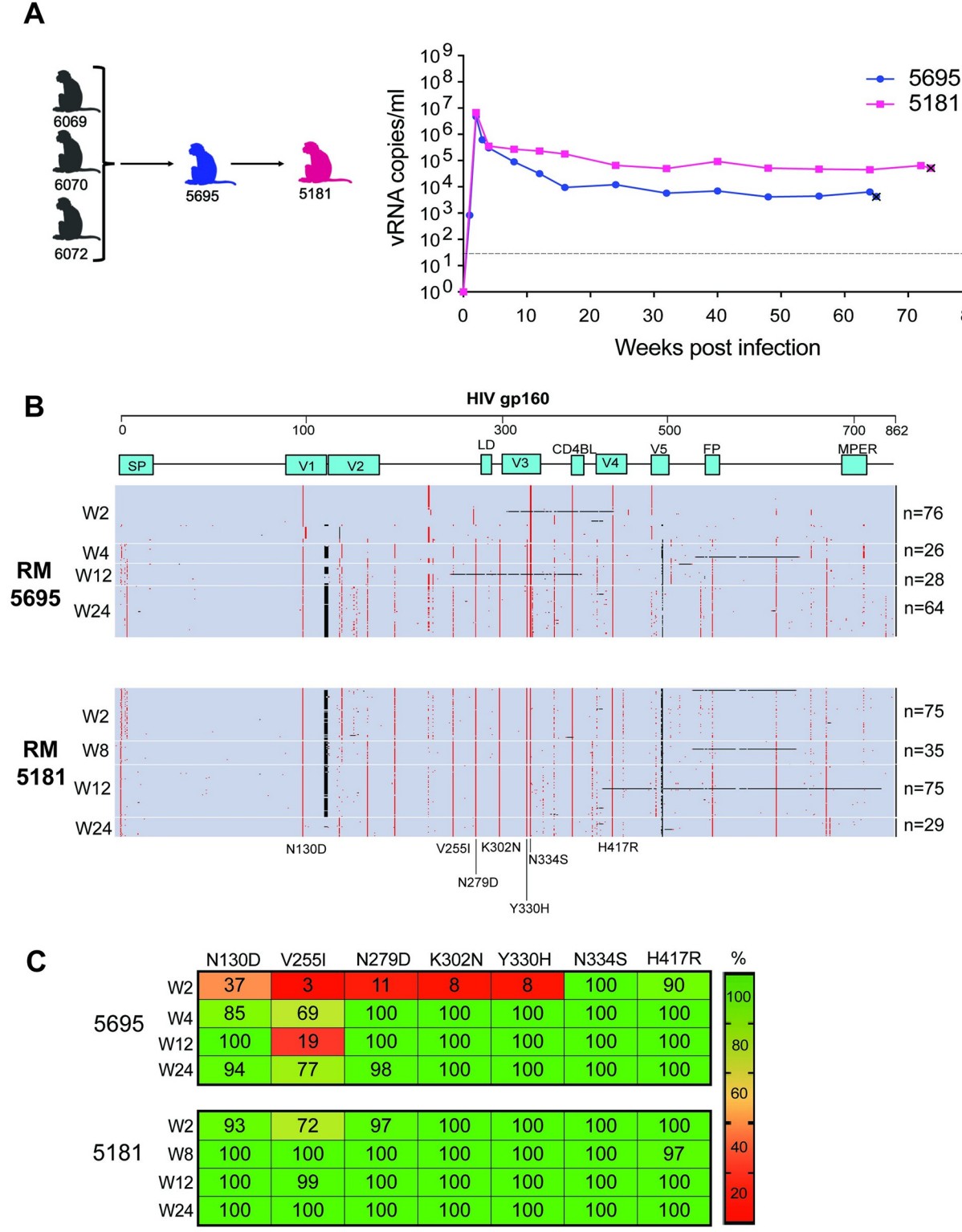

**Fig 3. *In vivo* competition of longitudinal plasmas from SHIV.C.CH505-infected RM.** (**A**) RM 5695 was intravenously inoculated with 1.4 mL plasma from three timepoints between weeks 4 and 20 from RM 6069, 6070, and 6072 (specifically, 200 ul each from week 4, 10, and 20 from RM 6072, weeks 10 and 20 from RM 6069, and weeks 10 and 20 from RM 6070). RM 5181 was inoculated with 0.8 mL of RM5695 plasma (200 uL each from weeks 4, 8, 12, and 24). Plasma viral load kinetics from RMs 5695 and 5181 are depicted. The dotted line indicates the assay's LoD (62 copies/ml). Euthanasia timepoints are marked with the x symbol. (**B**) Highlighter plot of SGS-derived gp160 Env sequences from

longitudinal plasma collected from RM 5695 and 5181. (**C**) Frequency of mutations in intravenously inoculated RM 5695 and 5181. Fig 3A was created with BioRender.com.

signature mutations outcompeting SHIV.C.CH505.TF. At week 4, all sampled sequences had four of the signature mutations (K302N, Y330H, N334S, and H417R) and most sequences had the remaining two mutations (N130D and N279D). By week 12 and 24, all six signature mutations were fully penetrant and persisted in all sequences (except for N130D and N279D, observed at a frequency of 94% and 98%, respectively, in RM 5695). Notably, while sequences continued to evolve, no other mutations arose to full penetrance by week 24 of infection (Fig 3C and S1 File).

In a second animal, RM 5181, a total of 800 μl plasma from RM 5695 (200 μl from 2, 4, 12, and 24 WPI) was IV administered. The RM 5181 inoculum was genetically heterogeneous and highly enriched for signature mutations that had become predominant early in RM 5695. The clinical course of RM 5181 was notable for even higher setpoint viremia ($>10^5$ c/ml), and disease progression necessitating euthanasia at 74 weeks of infection (Fig 3A). A total of 214 SGS-derived plasma *env* sequences were derived over four timepoints (median 55 per timepoint). Sequence analysis revealed early dominance of the signature mutations, with each mutation representing 93–100% of TF variants at week 2 and persisting at high levels longitudinally (Fig 3A and 3C). Thus, both RM 5695 and 5181 demonstrated selection for signature mutants with high-level viremia that persisted until clinical disease progression.

In addition to the previously identified signature mutations, a new mutation, V255I, arose in RM 5695 and became penetrant in RM 5181. The mutation V255I was detected in just one of the three RM plasmas that constituted the RM 5695 inoculum (44% of 6069 at 20 WPI) and rose to 59% frequency in RM 5695 at 6 months. In RM 5181, V255I represented 72% of week 2 viruses, rising to 100% prevalence within the population by week 8 and remaining penetrant through week 24.

## Signature mutation position and inferred function

To better understand the putative roles of the signature mutations, all seven residues (130, 255, 279, 302, 330, 334, 417) were modeled onto the CH505 SOSIP Env monomer, PDB ID 6VY2, and color-coded using the ACEMD software [55] (Fig 4). The relative conservation of gp160 residues across circulating HIV-1, characterized by Shannon Entropy, was calculated from a 208-virus panel of genetically and geographically diverse strains representing major virus subtypes and circulating recombinant forms. The most variable of the seven residues is position 130, which is highly exposed in the N-terminus of the V1 loop near the apex of the Env trimer. Position 130 is occupied by 12 different amino acids among the 208-virus panel, with 130D representing just ~9% of strains. All other positions of interest are more conserved, and align within or near the CD4 [56,57]) or coreceptor binding sites ([58]) (Fig 4A). Three signature mutations (K302N, Y330H, N334S) encode amino acids that are highly conserved (67–99%) [59] across HIV-1 sequences. Structurally, positions 255 and 417 lie within in the gp120 bridging sheet (255 is located in C2 and 417 is located in V4), which links the inner and outer domains of gp120 Env upon binding to CD4; position 279 lies within loop D of the CD4 binding site. These three positions (255, 279, and 417) are less variable (lower Shannon Entropy with only four to six distinct residues sampled at these positions globally), and the signature mutation residues are less commonly seen than the amino acid present in SHIV.C.CH505.TF (255 and 279), or neither mutation is common (417). Functionally, these Env residues (255, 279, 417) have been shown via mutagenesis studies to regulate binding and/or entry into CD4

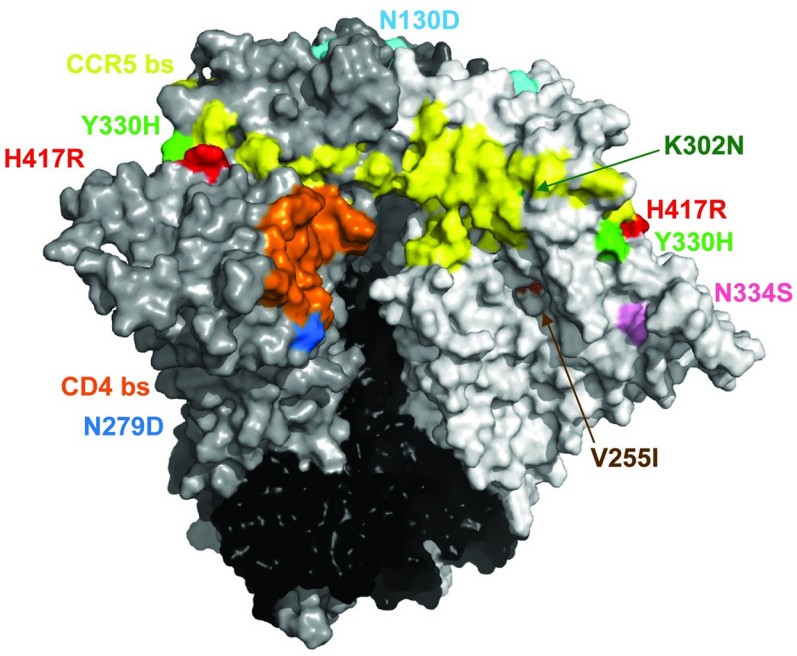

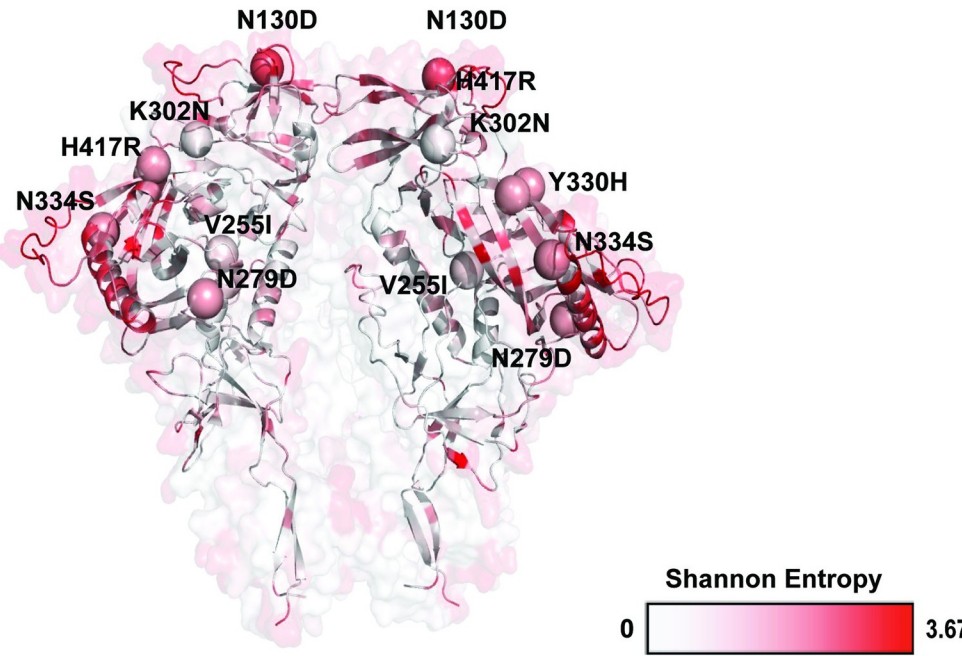

**Fig 4. Surface projection and conservation of signature mutations.** (**A**) The seven signature mutations are shown on a surface model of the non-glycosylated CH505 SOSIP Env trimer, PDB ID 6vy2. The CD4 binding site is highlighted in orange, the CCR5 binding site in yellow, and the seven mutations are color-coded, with K302N and V255I indicated with arrows, as they are largely hidden in this view. (**B**) Ribbon structure of CH505 SOSIP, with conservation shown by color and the seven mutations depicted as spheres. Conservation is defined by Shannon entropy, ranging from 0 to 3.67, as determined from a 208-virus panel [81].

and CCR5-expressing cells [57,59,60]. Positions 302, 330, and 334 are also structurally related to entry, lying adjacent to the CD4 binding site within C2 (302) or C3 (330, 334). Several mutations are also associated with escape from the early autologous nAb responses in SHIV.C. CH505-infected RM [41]; N130D, N279D, and N334S confer 3-fold escape from early nAb responses (week 10 plasma and V3-targeting mAbs), while K302N in isolation appears to modestly sensitize the virus to week 10 and week 20 plasma antibodies [41]. N334S encodes the shift of a glycan, which fills a glycan hole important in maturing bnAb development [61]. Together, previous studies, residue position, and Shannon entropy measures suggest that many of the mutations arose via escape from early immune responses or reversion to group M consensus and many have a potential influence on virus entry via CD4 or coreceptor binding.

### *In vitro* replication advantage of single and combination mutations

To determine the contribution of individual signature mutations to virus replication, full-length SHIV.C.CH505 infectious molecular clones (IMCs) containing each of the identified signature mutations (except N130D) in isolation were generated, shown to be infectious, and tested in parallel replication experiments on primary RM CD4 T cells. Fig 5B shows results from a representative experiment testing the replication of each single mutant clone and SHIV.C.CH505.TF over 9 days. Except for V255I, all signature mutations conferred a modest, non-significant increase in replication over wildtype SHIV.C.CH505 (median of 3-fold increase in p27 through day 6), p = 1, Dunnett's multiple comparisons test) across three independent experiments (Fig 5B and 5C). The IMC containing V255I, in contrast, replicated significantly less than SHIV.C.CH505.TF (median 29-fold decrease in p27, p<0.0001, Dunnett's multiple comparisons test, Fig 5B).

We next generated IMCs encoding combinations of the six signature mutations in a newer TF SHIV backbone, 3C, which contains deletions in the *tat* and gp41 *env* reading frames that were selected *in vivo* in multiple RM infected with distinct TF SHIVs (*eg*, SHIVs encoding clade B, C and D Envs), conferred a fitness advantage *in vitro* compared to the previous backbone [51], and has been incorporated into current iterations of TF SHIVs and all SHIVs described here moving forward [52]. Combination mutant clones (named according to the number of signature mutations present compared to SHIV.C.CH505.TF, see Fig 5A) encompassed both 4- and 5-mutant combinations observed *in vivo* as well as those lacking one or more specific mutations (3MA-3MD). When tested in three independent experiments on distinct donor RM CD4 T cells, the combination clones replicated substantially better than SHIV. C.CH505.TF (median 8 to 17-fold increase in p27 at day 3–6 of culture, Fig 5D and 5E), except for clone 3MA (median 0.2-fold decrease). Although three combination clones exhibited significant increases in replicative capacity compared to SHIV.C.CH505.TF across experiments (5MB, 3MB, and 3MD, p<0.05, Dunnett's multiple comparisons test), no single clone was clearly dominant across all donor cells. The roles of two specific mutations, however, are evident. From the modest enhancement in replication of K302N single mutant and the markedly reduced replication of the only combination mutant without K302N (3MA), we can infer 302N substantially contributes to improved replication of the adapted viruses. Finally, V255I alone significantly impairs replication but is neutral or advantageous when in combination with other potentially compensatory signature mutations.

### Enhanced replication due in part to entry mechanism

As the signature mutations lie within or proximal to CD4 and coreceptor binding sites, we asked whether enhanced replication could be attributed to more efficient virus entry. To assess the contribution of signature mutations to differences in cell entry, ZB5 cells, which stably

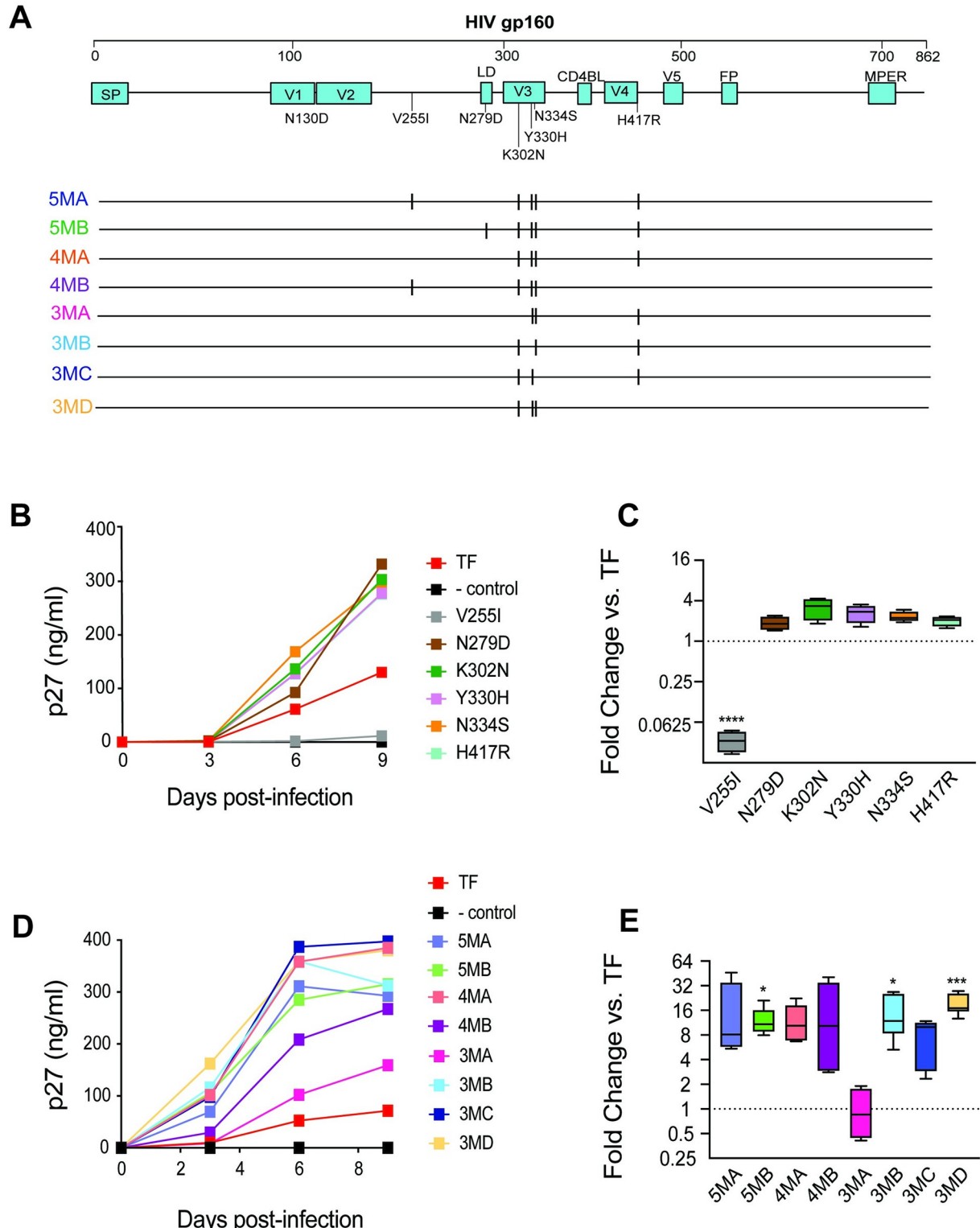

**Fig 5. *In vitro* replication of infectious HIV-1 molecular clones bearing signature mutations.** (A) The top panel represents the gp160 Env sequence of the SHIV.C.CH505.TF IMC depicted along with the positions of signal peptide (SP), variable loops (V1-V5), CD4 binding loop (CD4BL), fusion peptide (FP), membrane-proximal external region (MPER), and mutations N130D, V255I, N279D, K302N, Y330H, N334S, and H417R. IMCs containing different combinations of signature mutations are depicted underneath. (B) Replication of mutant viruses containing single mutations compared with SHIV.C.CH505.TF are shown in primary, activated RM CD4 T cells over 9 days of culture as

determined by supernatant p27 Ag levels (ng/ml). The negative control virus was Env-defective SHIV. (C) Fold-difference at peak replication vs. TF is shown. **** p < 0.0001—One-way ANOVA followed by Dunnett's multiple comparisons tests. (D) The replication of viral clones containing combinations of mutant alleles (5MA, 5MB, 4MA, 4MB, 3MA, 3MB, 3MC, 3MD) is shown compared with SHIV.C.CH505.TF in RM CD4 T cells over 9 days of culture as determined by supernatant p27 Ag levels (ng/ml). (E) Fold-difference at peak replication vs. TF is shown. * p ≤ 0.05, *** p ≤ 0.001- One-way ANOVA followed by Dunnett's multiple comparisons tests.

express rhesus CD4 and CCR5 [51] and encode a luciferase reporter, were inoculated with the 14 single and combination mutation clones, and assessed for virus entry. An additional clone with the N130D mutation in isolation was included as a comparator.

As shown in Fig 6, the Y330H mutation appeared to modestly improve entry into ZB5 cells in isolation, while N130D, N279D, N334S, and H417R conferred retained or minimally improved entry (Fig 6A). Two mutations (V255I and K302N) were significantly deleterious to entry in isolation (59 and 49% decrease in entry, respectively, p<0.05, Dunnett's multiple comparisons test) (Fig 6A). Among combination clones, three clones (5MA, 3MD, and 5MB) demonstrated significantly enhanced entry compared to the TF (>64% increase, p<0.0001 for 5MA and 3MD, and >45% increase, p<0.001 for 5MB, Dunnett's multiple comparisons test) (Fig 6B). Mutants lacking either K302N, Y330H, or N334S (3MA, 3MB and 3MC, respectively), had lower median entry efficiency (Fig 6B). Together, these findings demonstrate that improved entry is a likely mechanism for enhanced replication and that Y330H, as well as other V3 mutations, may drive the improved entry of combination mutants. Further, results suggest that no single mutation in isolation is wholly responsible for enhanced entry, and that key signature mutations (*eg*, V255I and K302N), impair entry when present without compensatory mutations.

## Env conformation of signature mutant clones

Primary HIV-1 isolates generally encode Envs that are neutralization resistant to most polyclonal anti-HIV antibodies from chronically infected humans. Such Envs have "closed" (non-CD4-triggered) trimer structures and have been termed tier 2 or tier 3 to distinguish them from "open" tier 1 viruses [39]. Many laboratory-adapted viruses, however, develop enhanced *in vitro* replication via more open, non-native Env conformations that reveal previously masked epitopes [60,61]. We looked for altered Env conformations—a marker of tier 1 neutralization sensitivity—that confer neutralization sensitivity using two mAbs; 3869, which targets linear V3 peptides that are conformationally buried and inaccessible in tier 2 viruses, and 17B, which preferentially binds CD4-induced epitopes in the bridging sheet that are expressed in CD4-triggered or otherwise more open trimers (Fig 6C–6F).

SHIV.C.CH505.TF, expressing a primary HIV-1 Env, is resistant to neutralization by both 3869 and 17B [39] (Fig 6C and 6D). The single mutant containing K302N has a markedly more open conformation, with potent neutralization by both 3869 and 17B. The remainder of the single mutant IMCs retain largely shielded conformations. In the combination IMCs, clones 5MA and 5MB, which represent mutational patterns commonly sampled *in vivo*, are fully resistant to both mAbs (Fig 6E and 6F). Clone 4MA, which lacks mutations at either C2 position (255 or 279), is more sensitive to both mAbs, though it does not reach 50% neutralization even at high concentrations. Clones 3MB and 3MC are more sensitive to these mAbs, with IC50s less than 1 μg/ml for both antibodies. These three clones (4MA, 3MB, 3MC) are the only ones that contain both K302N and H417R without either N279D or V255I, suggesting that these C2 mutations may function to close Env conformation altered by V3 and V4 mutations. Together, these data suggest that several of the V3/V4 signature mutations, which drive

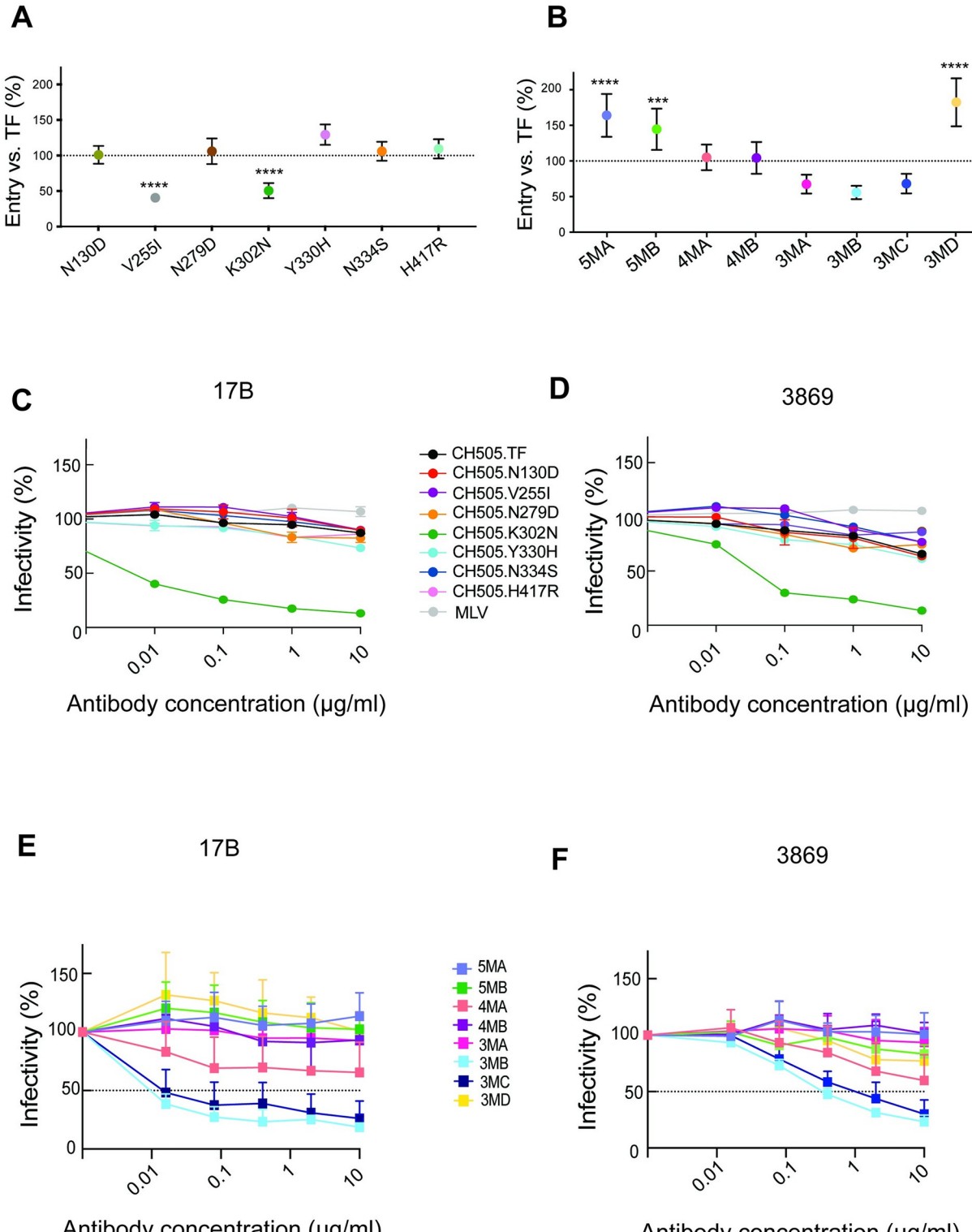

**Fig 6. Cell entry efficiency and Env conformation of single and combination mutant SHIV.C.CH505 IMCs. (A, B)**. Entry efficiency by single and combination mutant clones was evaluated in a single cycle entry assay in ZB5 cells that express rhesus CD4 and CCR5 [82,83]. **(C-F)** Env trimer conformation or "openness" was assayed by measuring virus neutralization sensitivity to anti-HIV mAbs that target linear V3 (3869) or CD4-induced bridging sheet (17b) epitopes. The single mutant K302N and the combination mutants 3MB, 3MC and 4MA are "open" compared with the closed structure of the TF CH505 Env and other mutants. Values are mean (SD); *** p ≤ 0.001, **** p < 0.0001—One-way ANOVA followed by Dunnett's multiple comparisons test.

improved *in vitro* entry and/or replication, alter protein conformations that require compensatory C2 mutations.

## *In vivo* competition of candidate SHIV.C.CH505 combination clones

To better characterize virus fitness over multiple replication cycles, we performed an *in vivo* competition experiment using a lower-dose inoculum to enable direct competition between the combination mutant variants. We sequenced frequently over the first 4 weeks of infection, where exponential growth, abundant target cells, and non-existent or early nascent adaptive immune responses allow a more direct assessment of replication capacity [59]. Three outbred RM were infected with 80 ng p27 of an inoculum stock comprised of equal amounts of each variant (10 ng p27 Ag of each variant; 80 ng *in toto*) (Fig 7A) and followed longitudinally. All animals became productively infected with peak VL ranging from $10^4$ to $10^6$ c/ml at week 2 (Fig 7B). In all RM, viremia was maintained above $10^2$ c/ml through 48 WPI, with set point VLs ranging from $10^{3-}10^5$ c/ml through more than a year of infection.

To assess the relative fitness of signature mutants *in vivo*, we employed a deep sequencing strategy with amplicons spanning a 500-nucleotide region in Env encompassing all candidate mutations using the Ilumina platform, generating sequences for the inoculum and plasma *env* sequences from 1, 2, and 4 WPI (Fig 7). In parallel, SGS of gp160 *env* for weeks 1 and 2 post-infection was also performed (S2 Fig). SGS precludes *in vitro* recombination but has relatively limited power to detect minor variants unless large numbers of sequences are obtained. Illumina sequencing allows for greater sequencing depth but can induce substantial *in vitro* recombination from a bulk PCR step. Thus, we used the SGS to screen for evidence of *in vivo* recombination. SGS *env* sequences from weeks 1 and 2 (n = 162) identified all eight inoculum variants with no evidence of *in vivo* recombinants (S2 Fig); thus, we excluded the possibility of high frequency *in vivo* recombination, and excluded *in vitro* recombinant NGS sequences from further analyses.

Illumina sequencing of the inoculum stock confirmed the presence of all 8 variants. Sequences from 1 WPI demonstrated that all 8 variants were transmitted and replicated systemically in each RM. Notably, variant 5MA rapidly outcompeted all others. 5MA represented the largest share of plasma variants at all timepoints across the three RM, comprising 76 and 99% of plasma variants on average by week 2 and week 4, respectively. Variant 4MB, which lacks only H417R compared to 5MA, was the only other variant to persist at over 10% viruses in any animal at week 4. Variant 4MA, which lacks only V255I compared to 5MA, comprised the largest share of the inoculum, and remained detectable at low frequency at week 4 across animals. Thus, with the substantial *in vivo* advantage of clone 5MA, its closed Env conformation (Fig 6E and 6F), and *in vitro* replication advantage (Fig 5D), we moved forward with this as the most promising variant, naming it SHIV.C.CH505.v2.

## SHIV.C.CH505.TF vs. SHIV.C.CH505.v2 *in vivo* competition

To confirm and quantify the fitness advantage of SHIV.C.CH505.v2 (variant 5MA), we performed a direct *in vivo* competition between this variant and SHIV.C.CH505.TF. Four RM were inoculated with a stock containing equal amounts of TF and V2 viruses as determined by p27 antigen measurement. Two RM (T276 and T277) were intravenously challenged with a lower dose (15 ng p27 Ag of each variant, 30 ng *in toto*) and two RM (T278 and T279) were challenged with a higher dose (150 ng p27 Ag of each variant, 300 ng *in toto*) (Fig 8A). All RM became productively infected with peak VL of $10^{5-}10^6$ c/ml within 2 WPI (Fig 8B). In all four RM, viremia was maintained in the $10^{3-}10^5$ c/ml range through more than a year of infection.

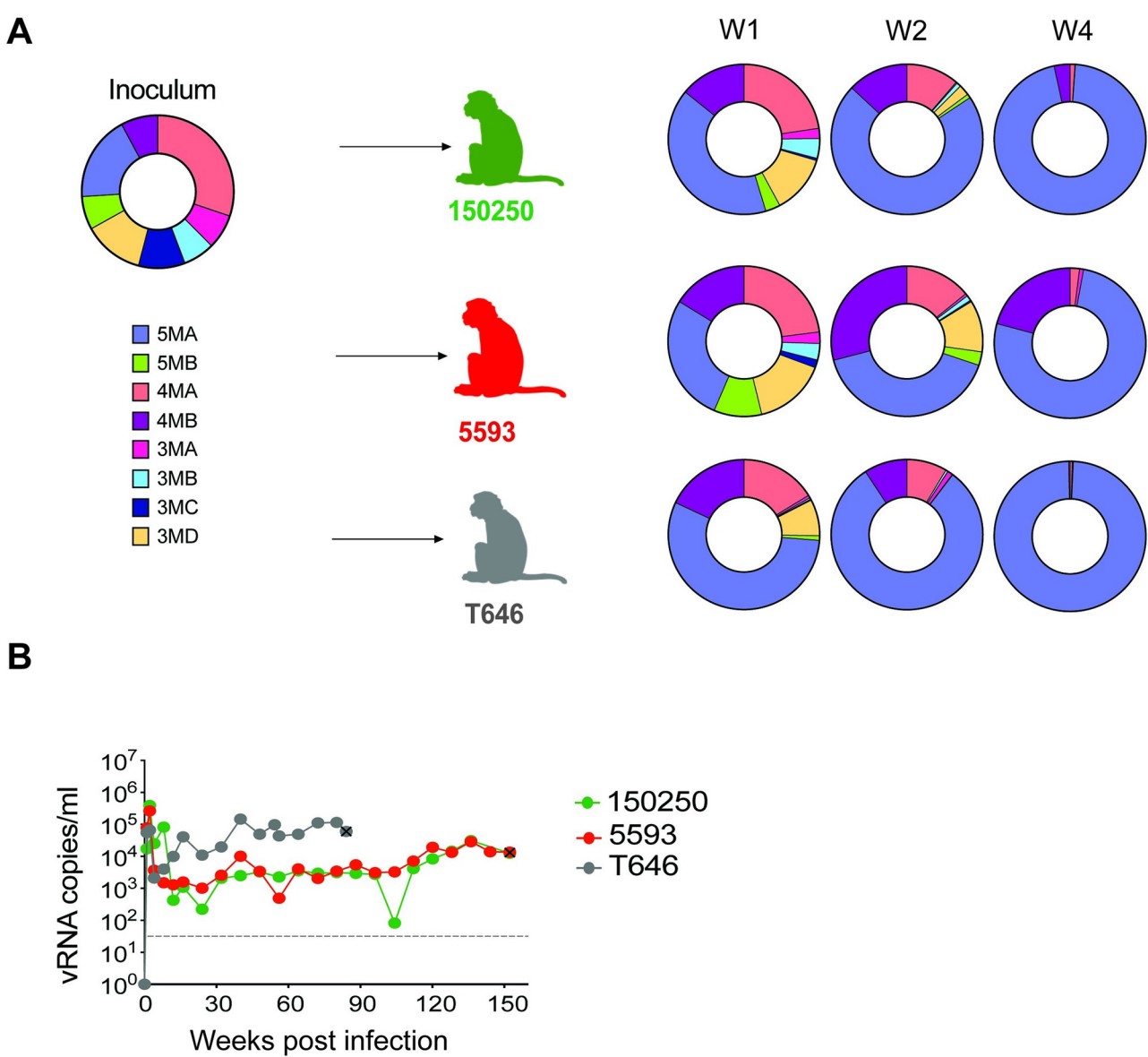

**Fig 7. *In vivo* competition of candidate SHIV.C.CH505 combination clones.** Three outbred RM were infected with 80 ng p27 Ag of an inoculum stock comprised of approximately equal amounts of 8 variants (**A**, left) and followed longitudinally for plasma virus load (**B**) and for relative proportions of each variant at weeks 1, 2, and 4 post-infection as determined by Illumina sequencing and shown in the donut plots (**A**, right). Mutant 5MA replicated most efficiently in all three animals. In this experiment, the wildtype TF SHIV.C.CH505 was not tested to allow for a more direct comparison of the replication efficiency of the different combination mutants. Fig 7A was created with BioRender.com.

Deep sequencing of the region encompassing the 5 signature mutations and SGS of gp160 *env* of the inoculum stock were used to assess variant frequencies in the inoculum and over the first four weeks of infection. Despite our attempt to generate a stock with equal components of the two variants, SGS and deep sequencing revealed the inoculum stock was comprised of between 60–72% SHIV.C.CH505.v2 and analyses were adjusted accordingly. The proportions of the two variants were then tracked over the first 4 WPI (Fig 8). To overcome ambiguity from *in vitro* recombination of the deep sequencing process, we focused on the frequency of TF (V) and V2 (I) amino acids at position 255, as this mutation was the least likely to be

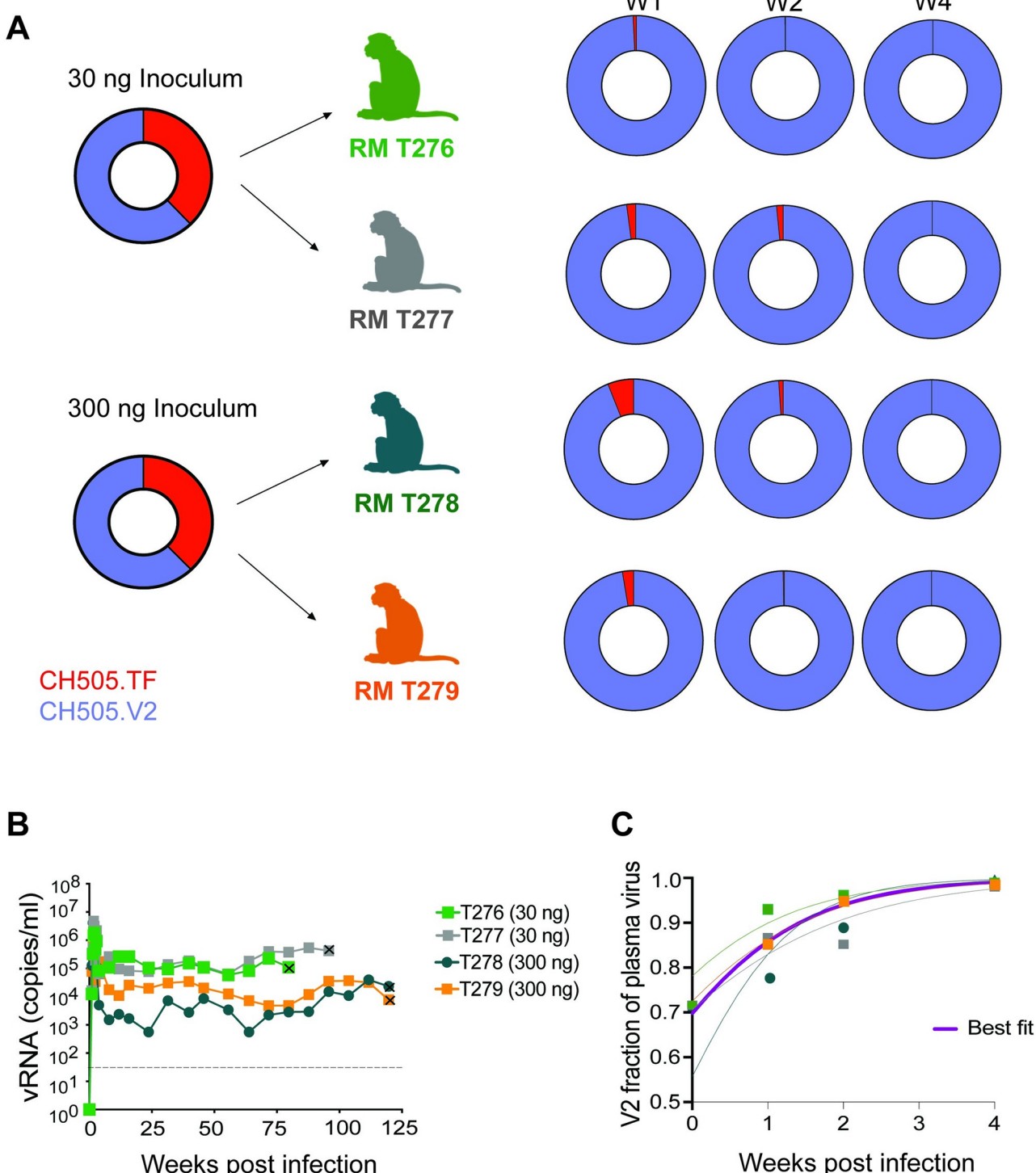

**Fig 8.** *In vivo* **competition of SHIV.C.CH505.v2 vs. SHIV.C.CH505.TF.** (**A**) Four RM were IV inoculated with approximately equal amounts of SHIV.C.CH505.v2 (variant 5MA) and SHIV.C.CH505.TF. Two animals (RM T276 and T277) received 15 ng p27 Ag of each variant whereas two other animals (RM T278, and T279) received 150 ng p27 Ag of each variant. The relative proportion of the different viruses in the challenge stock and at weeks 1, 2 and 4 post-infection as determined by illumina sequencing are shown in donut plots. (**B**) *In vivo* plasma virus kinetics are shown over time. The dotted line indicates the assay's LoD (62 copies/ml). Euthanasia timepoints are marked with the x symbol (**C**) Modeling of the relative frequency of V2 vs. TF in the 4 RM over time is shown with a best-fit approximation in purple. Fig 8A was created with BioRender.com.

selected for in isolation in the TF *in vivo* given the fitness cost it conferred (Fig 4C). Comparing the virus populations over time, we observed enhanced growth rates of V2 vs. the TF, with V2 frequencies of between 85.2 and 96.1% of plasma virus by 2 WPI, and 98.0 and 99.8% by 4 WPI (Fig 8B). Modeling the relative growth rates of the TF and V2 variants, we found that SHIV.C.CH505.V2 had an average estimated growth advantage of 0.14 days-$^1$ (range: max = 0.19 $days^{-1}$, min = 0.10 $days^{-1}$) (Fig 8C). This differential in growth rates predicts a doubling time for the ratio of V2 to TF of just 5 days. Thus, in the first 4 WPI, the V2:TF ratio doubled more than 5 times to comprise the vast majority of replicating virus.

To assess the relative fitness of these viruses over a longer timescale, we sequenced available longitudinal samples through 32 weeks of infection. Given concern for recombination within the diversifying virus pool, we performed SGS of gp160 *env* and report the relative frequency of each of the five signature mutations, as well as N130D and N279D (S3 Fig). Excluding sequences from week 1, we see full penetrance of four of the five signature mutations (K302N, Y330H, N334S, and H417R) in all RM at all timepoints. At position 255, we observed a low fraction of back mutation sporadically in RM T276 and an increased frequency in RM T277 at later timepoints. Notably, back mutation at position 255 was often associated with presence of N279D. Together, longitudinal *env* sequencing confirms the relative fitness advantage of SHIV.C.CH505.v2 over SHIV.C.CH505.TF, and reinforces a likely shared role of the V2 mutations (V255I and N279D) in compensating for the V3 mutations (S3 Fig).

## Epitope mapping of SHIV.C.CH505.v2

Next, we performed a more thorough comparison of SHIV.C.CH505.TF vs. SHIV.C.CH505. v2 sensitivities to neutralization by clinically relevant bnAbs (S4 Fig). In many respects, these two viruses remain similar, but sensitivities to key epitopes are predictably shifted. First, the addition of the N332 glycan renders SHIV.C.CH505.v2 highly sensitive to V3 glycan supersite targeting bnAbs, including PGT121 and 10–1074 [62]. A previous study found that Env residue 255 substitution results in decreased binding of HIV-1 isolates to soluble human CD4 and increased sensitivity to V2 apex bNAbs [60]; and mutations in CD4 binding site and bridging sheet are also predicted to alter CD4bs-targeting bnAb sensitivities [63]. In the TZM.bl assay, SHIV.C.CH505.v2 demonstrates modestly decreased sensitivity to CD4 binding site bnAbs, but retains reasonable sensitivity to more potent CD4bs bnAbs (*e.g.*, VRC07.523LS, N6). SHIV.C.CH505.v2 also has modestly increased (3-fold) sensitivity to some V2 apex bNAbs, including CAP256 (S4 Fig). Together, SHIV.C.CH505.v2 has a desirable phenotype for bnAb studies, as it retains substantial sensitivity to CD4bs bnAbs and V2 Apex bnAbs, and gains sensitivity to V3 glycan bnAbs.

## *In vivo* kinetics and persistence of SHIV.C.CH505.v2

Finally, we tested SHIV.C.CH505.v2 infection kinetics over time and persistence through treatment with suppressive antiretroviral therapy (ART). Four outbred RM were each IV inoculated with 300ng p27 of a barcoded SHIV.C.CH505.v2. The barcode approach was adapted from the work of Keele, Davenport and colleagues [64] and incorporates a cassette with a random 10 nucleotide sequence between Vpx and Vpr, enabling enumeration and tracking of variants over time and tissues [65,66]. All four RM were productively infected, with peak viremia between $10^6$–$10^8$ c/ml, and setpoint viremia between $10^3$–$10^6$ c/ml (Fig 9). Consistent with a high dose IV inoculation, barcode sequencing of peak viremia confirmed a high multiplicity of infection, with a median of 1,202 (range 563 to 2,112) unique barcoded viruses, or clonotypes, replicating systemically across the four RM. RM remained viremic off ART for 9 months, then started on a 6-month course of ART (subcutaneously daily administered dolutegravir,

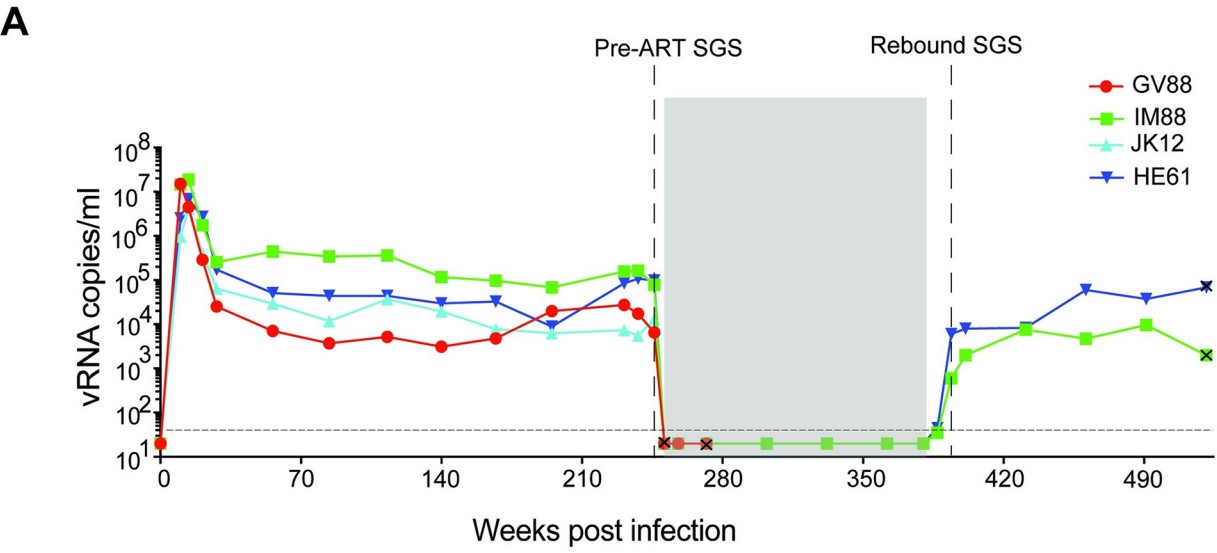

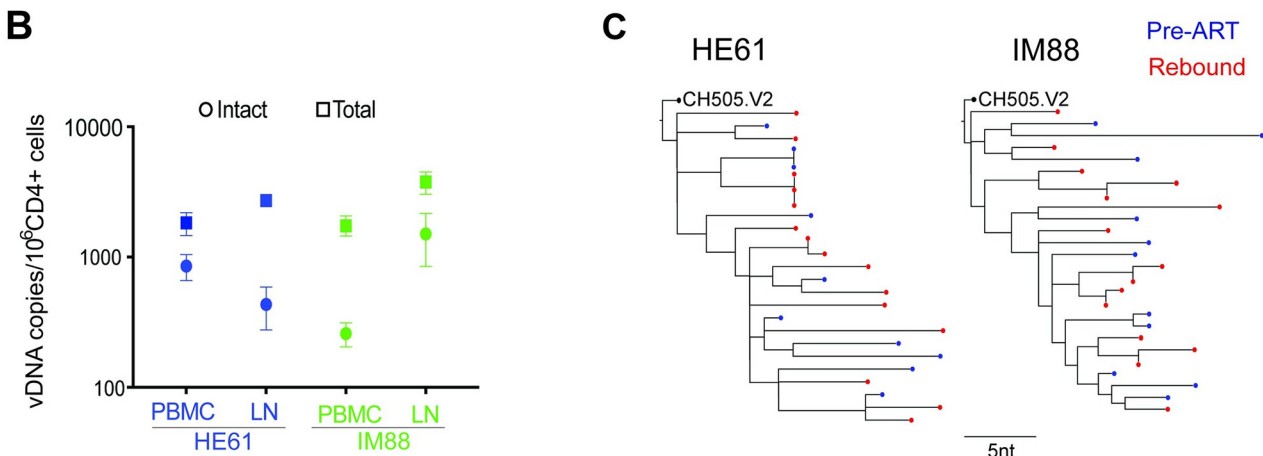

**Fig 9. SHIV.C.CH505.v2 persists through suppressive ART.** (**A**) *In vivo* viral kinetics of 4 RM IV infected with barcoded SHIV.C.CH505.v2, with daily ART initiated at 9 months post-infection, maintained for 5 months, then monitored for an additional 6 months post-ATI. The grey box indicates the 5-month period of daily antiretroviral therapy (dolutegravir, emtricitabine and tenofovir) administered subcutaneously. (**B**) Intact and total proviral DNA levels for two RM sampled on suppressive ART at day 330 (PBMC) and day 358 (LN) post-infection. (**C**) SGS of pre-ART and rebound plasma *env* reveals multiple distinct rebound virus lineages reactivating from latency.

emtricitabine, and tenofovir). RM suppressed virus below the limit of assay detection within 2 weeks and remained suppressed throughout ART. Unfortunately, two RM experienced wasting near the end of viremia, which continued through early ART suppression, and thus had to be euthanized prior to treatment interruption (day 265–275 post-infection). For the remaining two RM, measures of the intact and total proviral DNA reservoir in PBMCs and peripheral lymph node mononuclear cells on ART (Fig 9B) demonstrated SHIV persistence, and interruption of ART after 5 months of suppression led to rapid virus reactivation with low-level detectable viremia at 2 weeks and VL > 1,000 c/mL by 4 weeks post-ATI, followed by return to near pre-ART setpoint viremia (Fig 9A). SGS of pre-ART and rebound plasma gp160 *env*

revealed multiple distinct rebound virus lineages reactivated from latency to replicate systemically, reflecting a diverse virus reservoir that persisted through suppressive ART (Fig 9C). Acknowledging the limited number of animals in this experiment, with just four SHIV.C. CH5050.v2-infected RM with ~1 year of infection and just two with persistence through durable ART suppression, the available SHIV.C.CH505-infected RM recapitulate key components of HIV-1 infection, including kinetics and clonality of ART treatment and interruption.

## Discussion

While there is no ideal virus for all experiments conducted in the NHP model of HIV-1 infection, certain viral characteristics are essential for most applications. A useful viral reagent should generally encode a primary HIV-1 Env with a closed conformation and tier 2 neutralization resistance, be transmissible via parenteral and mucosal routes, infect appropriate target cells, exhibit reliable viral replication kinetics, lead to HIV-like pathogenesis and immunobiology, display relevant bnAb epitopes, and engage innate and adaptive immune responses akin to HIV-1 infection. For experiments of persistence or cure interventions, the virus additionally should produce consistent viremia and immunopathogenesis over time and respond to antiretroviral therapy with suppressed viremia and establishment of persistent reservoirs that reactivate upon treatment interruption. In this study, we derived a virus, SHIV.C.CH505.v2, that fulfills these criteria.

Based on the discovery of conserved patterns of virus evolution in multiple SHIV.C. CH505-infected RM, we identified a signature of mutations associated with high virus replication. *In vitro* analyses delineated the effects of individual or combinations of mutations on virus replication, entry, and neutralization sensitivity. We then leveraged *in vivo* competition experiments to determine the viral variant with optimal overall fitness across multiple outbred animals. Together, these approaches identified a variant, SHIV.C.CH505.v2, which differs by five amino acids from SHIV.C.CH505.TF but has substantially improved *in vivo* replication fitness and preserved primary HIV-1 Env antigenic properties.

Combining sequence analysis of SHIV.C.CH505-infected RM, results from *in vitro* entry, replication, and neutralization assays, as well as *in vivo* competition experiments, we could discern the phenotypic effects of individual signature mutations. Four of the commonly seen mutations (N130D, N279D, N334S, and H417R) were previously shown to arise in response to early adaptive immune responses. The first three confer escape from early autologous nAb responses targeting a CD4bs-like epitope in SHIV.C.CH505-infected RM [41], while the last confers escape from early CD8 T cell responses. N130D, located in variable loop 1, consistently arises early in SHIV.C.CH505-infected RM, but often reverts to asparagine in replicating virus populations at later timepoints, likely as a consequence of autologous neutralizing antibody pressure [41]. The longitudinal patterns of N130 diversification in SHIV.C.CH505-infected RM and in global virus strains (Fig 4) support this interpretation. The N279D mutation in loop D, similarly, arises co-incident with autologous antibody responses that target the CD4bs, and this mutation has been shown to impair binding of several classes of CD4bs antibodies [63]. This mutation persists in most SHIV.C.CH505-infected RM, with entry and replication experiments suggesting minimal impact in virus fitness. Thus, we inferred that N130D and N279D represented escape mutations in RMs 6072, 6069 and 6070, and because they were not present in the best replicating variants with combinations of mutation *in vivo*, they were not included in final design of SHIV.C.CH505.v2.

The mutation N334S arises early, shifts the N-linked glycan from residue 334 to 332, and remains at full penetrance in nearly all infected animals (Figs 1 and 3) [41,50,67]. This, plus the fact that S334 represents the global consensus suggests it confers a substantial fitness advantage

over the wildtype allele. This fitness advantage is accentuated in SHIV.C.CH505 infected animals that commonly target a nearby glycan hole when the N-linked asparagine is present at residue 334 instead of 332. H417R, which lies in the C-terminus of V4, is also a reversion to group M consensus that has been shown to be a CTL escape reversion [40,41]. This mutation arises early, persists in most infected RM, and confers a modestly increased *in vitro* replication rate without substantial change in entry efficiency. Thus, two of the identified mutations (N334S, H417R) that arose as escape mutations from adaptive immune responses in the donor subject to CH505, conferred fitness benefits and were included in SHIV.C.CH505.v2.

The two mutations with the most substantial evidence for enhancing fitness are K302N at the N-terminus of V3 and Y330H at the C-terminus of V3. Both mutations arise early, persist nearly universally across infected RM, and are conserved within group M sequences [68]. Interestingly, N330 is among the most highly conserved sites globally among the six signature mutations that we identified, and K302N is responsible for marked replication improvements, as shown by the K302N single mutant's enhanced replication (Fig 5) and the severely impaired virus entry (Fig 6A), *in vitro* replication (Fig 5E), and *in vivo* growth of the combination mutant lacking K302N (Fig 7). This mutation does not enhance virus entry in isolation but instead either functions in combination with other mutations or through post-entry mechanisms. Notably, K302N disrupts the closed Env conformation when added to the TF in isolation or with certain other mutations (Fig 6), exposing both CD4-induced and linear V3 epitopes, but is not associated with an open conformation in combination with the complete set of mutations in SHIV.C.CH505.v2. Unlike K302N, Y330H conferred a modest increase in both replication fitness and cell entry (Fig 6).

The mutation at position 255, which is highly conserved globally and lies within the C2 region of the bridging sheet, is notable for several reasons. V255I arises late or infrequently in infected RM, suggesting several compensatory mutations are required for this mutation to have a selective advantage and retain viral fitness. In isolation, this mutation renders the TF nearly non-infectious (Fig 6), though it does not impair viruses containing at least three of the other key signature mutations (K302N, Y330H, N334S). Neutralization experiments (Fig 6) suggest that V255I, like N279D, rescues viruses with these V3/V4 mutations, returning the Env to a closed conformation [69,70]. *In vivo* competition experiments demonstrate the marked superiority of viruses containing V255I with the three V3 mutations (K302N, Y330H, N334S) across RM (Fig 7). The striking advantage shown by these V255I-containing mutants was not discernable *in vitro*, highlighting the value of *in vivo* competition in outbred animals to holistically assess multiple interrelated determinants of fitness.

Together, these results suggest that the five mutations within the most successful signature mutant, SHIV.C.CH505.v2, work cooperatively to enhance virus replication, at least partially through improved virus entry, while maintaining a closed Env conformation required to shield against polyclonal autologous neutralizing antibody responses. The resultant virus is striking in its *in vivo* advantage compared to the TF SHIV.C.CH505, with modeling quantifying a substantial growth advantage of 0.14/days$^{-1}$ and doubling time of V2:TF of 5 days. Across the eleven RMs infected with SHIV.C.CH505.v2 or swarms with substantial proportions of this variant, we found consistently high setpoint viral loads and no examples of spontaneous control. Indeed, in experiments comparing RMs inoculated with mixtures leading to different timing of SHIV.C.CH505.v2 predominance (Figs 3 and 8), we see a trend towards higher set point viremia with greater proportions of SHIV.C.CH505.v2 replication at early timepoints. This confirms the enhanced replicative fitness of SHIV.C.CH505.v2 and reinforces the importance of early virus-host interactions in longitudinal outcomes.

This study has limitations. The number of animals tested in each *in vivo* experiment is small. Further experiments using greater numbers of RM are needed to verify the viral

kinetics and persistence of SHIV.C.CH505.v2. Overall, results suggest that SHIV.C.CH505. v2 has many properties that recapitulate HIV-1. Several characteristics, however, such as rapid time to virus suppression on ART and lower setpoint viremia in a fraction of RM suggest less robust infection. These factors will need to be considered if investigators employ this virus in future studies characterizing reservoirs or testing cure interventions. Another limitation of the current study is that no UMI (unique molecular identifiers) were used in the next generation sequencing approaches. To support our inference of relative variant frequency, we used a standardize amount of quantified cDNA template for each sequenced sample according to sequencing depth. Further, we performed multiple experimental and technical replicates of each sample/timepoint sequenced. For example, to model the change in frequency between V2 and TF variants over time, we performed multiple sequencing runs on each sample, across multiple timepoints, for each of the four RM. Results across technical replicates, related timepoints, and animals showed the same trends, without any outlier or unexpected results. Similarly, the NGS and SGS results were supportive when performed on the same timepoints.

Desirable viral kinetics, paired with the extensively characterized immunobiology and pathogenesis of SHIV.C.CH505.TF, make SHIV.C.CH505.v2 a valuable tool in NHP models of HIV. The viral kinetics demonstrated by SHIV.C.CH505.v2-infected RM, with setpoint viremia in the $10^4$ to $>10^5$ copies/ ml range and persistence through suppressive ART, recapitulate the biology of HIV-1 infection. SHIV.C.CH505.v2 thus serves as an attractive NHP challenge virus that is more robust than several commonly used SHIVs, but less virulent than SIVmac239, which confers high-level viremia and rapid disease progression that exceeds the range experienced by people living with HIV-1 prior to ART. Further, SHIV.C.CH505.v2 has desirable sensitivity patterns to several classes of bnAbs in clinical development, including those targeting CD4 binding site, V3 glycan, and V2 apex epitopes. Given these attributes, SHIV.C. CH505.v2 has significant potential in NHP experiments characterizing HIV-1 prevention, pathogenesis, and cure.

## Material and methods

### Ethics statement

All experiments involving animals were approved by the Penn Institutional Animal Care and Use Committee (IACUC) and performed in compliance with the Animal Welfare Act and National Institute of Health Guidelines.

### Nonhuman primates

Indian-origin RMs were housed and cared at Bioqual LLC or Tulane National Primate Research Center according to the standards of the Association for the Assessment and Accreditation of Laboratory Animal Care International (AAALAC). All animals were tested and found to be negative for described SIV controller alleles *Mamu-A*01, B*08, and B*17*. Animals were sedated for SHIV inoculations, peripheral blood draws, and biopsies, as previously described [51]. Whole blood from animals was processed using centrifugation as described in [51]. PBMCs were isolated using Ficoll-Paque (GE Healthcare) gradient centrifugation. Plasma was clarified by centrifugation for 15 min at 3000 rpm, frozen, thawed, and then subject to viral RNA extraction as previously described [39]. SHIV VL levels in plasma were determined by quantitative real-time reverse-transcription PCR of SIV RNA by the Duke University IQVAC Core Laboratory as previously described [39].

## Construction and characterization of point and combination SHIV.C. CH505 clones

SHIV.C.CH505 clones containing signature mutations in isolation were generated and viral stocks were made via transfection of 293T cells as described. For the experiments described in Fig 5B and 5C, the SHIV.C.CH505.TF IMCs were generated in the original TF SHIV IMC [39], while in all other in vitro and in vivo experiments, all SHIVs (including SHIV.C.CH505. TF, SHIV.C.CH505.v2, and all combination mutants) were generated in the 3C backbone [52]. Briefly, amino acid change-conferring substitutions were placed into the original or 3C SHIV. C.CH505.TF using the Q5 Site-Directed Mutagenesis Kit (New England Biolabs #E0554S). Primers were designed according to the NEBaseChanger tool (https://nebasechanger.neb. com). All clones were sequenced in their entirety to verify that only desired mutations were present in the constructs. 293T cells were obtained from ATCC (# CRL-3216) and tested for mycoplasma contamination. 6 million cells were seeded in 10 ml of complete media (DMEM High Glucose Thermo Fischer Scientific #11965092) supplemented with Penicillin-Streptomy-cin-Glutamine (Thermo Fischer Scientific #10378016) and 10% Fetal Bovine Serum (Thermo Fisher Scientific #A5256701) for 24 hours at 37˚C, 5% $CO_2$. Transfection was achieved by add-ing 6 μg of plasmid DNA in 18 microliters of FuGENE 6 Transfection Reagent (Promega # E2691); the obtained mixture was incubated in plain media for 20 minutes and then added to the pre-seeded culture dishes. Cells were then incubated for 48 hours at 37˚C, at 5% $CO_2$. After 48 hours, supernatants containing virus were centrifuged at 3000 rpm for 8 minutes to pellet cellular debris and then aliquoted. Supernatants were not subjected to more than one freeze-thaw cycle.

The virion content of viral stocks was quantified using the Simian Immunodeficiency Virus (SIV) p27 Antigen Elisa kit from Zeptometrix (# 0801169) following the manufacturer's recommendations.

Titration on TZM-bl cells was performed as previously described [51]; briefly, TZM-bl cells (obtained from the NIH HIV Reagent Program (#ARP-8129) were seeded in a flat-bottom 96-well tissue culture plate at the density of 15,000 cells per well and incubated at 37˚C, 5% $CO_2$ for 24 hours. The following day, 5-fold dilutions of virus stock were plated, beginning with a virus stock dilution of 1:5 in complete DMEM media as mentioned above supplemented with DEAE-Dextran (Sigma Aldrich #30461) at the final concentration of 40 mg /ml; cells were then incubated at 37˚C, 5% $CO_2$ for 48 hours. At the end of the incubation step, media was removed, and cells were fixed by using a fixing solution prepared by adding 4 ml of glutar-aldehyde and 11 ml of formaldehyde to 500 ml of Dulbecco PBS (Thermo Fisher Scientific #14190144). After 10 minutes, cells were washed 3 times with Dulbecco PBS and stained by using a staining solution containing 1 mg/ml X-Gal, 5 mM potassium ferricyanide, 5 mM potassium ferrocyanide and 2 mM MgCI2. After 3 hours, cells were washed 3 times with Dul-becco PBS and the number of infected cells in each well was counted using the ImmunoSpot Anlayzer (S6 Ultimate M2 ImmunoSpot). The average number of positive cells in two conse-cutive dilutions across all 4 quadruplicates was determined and standardized to the input virus volume; the final titer was the mean of 2 consecutive titers.

Stocks of SHIV.C.CH505 clones containing signature mutations in combination were gen-erated as previously described [51]. Briefly, 8 candidate Envs containing variations of signature mutations were synthesized by Synbio Technologies and were cloned into the SHIV.3C back-bone using the BsmBI restriction sites at the 5' and 3' end of the CH505 Env cassette and then ligated together using T4 ligase (New England Biolabs #M0202S). 8 plasmids encoding full-length SHIV.C.CH505 combination clones were used to transfect 293T cells as described above. Infectivity on TZM-bl cells and virion content was determined as described above. The

infectivity of CH505 combination clone stocks ranged between $1.18 \times 10^{-3}$ and $9.00 \times 10^{-4}$ IU/particle, respectively, as determined on TZM-bl cells.

## Generation of SHIV.C.CH505.v2 and SHIV.C.CH505.TF barcoded stocks

Barcoded stocks of SHIV.C.CH505.v2 and SHIV.C.CH505.TF were generated as previously described [32,51]. Briefly, the CH505.TF and CH505.v2 Envs were cloned into a SHIV.3C backbone containing NotI at the barcode insertion site using the BsmBI restriction sites at the 5' and 3' end of the Env cassette and SHIV.3C backbone and then ligated together using T4 ligase. The barcode inserts consisted of single-stranded forward and reverse barcoded templates consisting of 10 random bases each. The forward and reverse barcode primers were heated to 95°C and then cooled at a rate of 1.5°C/min to enable annealing. Barcoded stocks of SHIV.C.CH505.TF and SHIV.C.CH505.v2 were digested with NotI. DNA was analyzed on an agarose gel (1%) and purified by using the QIAquick gel extraction kit (Qiagen #28706). DNA was then digested and primer dimers were ligated at 16°C overnight. The ligated, barcoded viruses were transformed into Max Efficiency Stbl2 Competent Cells (Thermo Fisher Scientific #10268019). The plasmid library was DNA extracted and barcodes were sequenced via the Illumina platform.

Infectivity on TZM-bl and virion content was determined as described above. The infectivity of barcoded SHIV.C.CH505.TF and SHIV.C.CH505.v2 stocks on TZM-bl cells were 1.86 and $4.3 \times 10^{-4}$ IU/particle, respectively. To assess the infectivity of SHIV.C.CH505.TF and SHIV.C.CH505.v2 on primary rhesus CD4 T cells, activated cells were plated at a density of $3 \times 10^5$ cells/well in 96-well flat-bottom tissue culture plates in RPMI 1640 Media (Thermo Fisher Scientific #11875199) supplemented with 15% Fetal Bovine Serum (Thermo Fisher Scientific #A5256701), 100 U/ml penicillin-streptomycin (Thermo Fisher Scientific #15070063), 30U/ml IL-2 (Prometheus Laboratories), and 40 µg/mL DEAE–dextran (Sigma Aldrich). 5-fold dilutions were plated, beginning with a virus stock dilution of 1:5; the virus stock in each well was diluted to a total of 60 µl of complete media. The virus mixture was incubated for 3 h at 37°C. Complete media was added to bring the total volume to 250 µl in each well and plates were incubated at 37°C, 5% $CO_2$. p27 antigen measurements were made on day 4 and $TCID_{50}$ was calculated by using the Reed Muench method [71]. The infectivity of viral stocks was determined via titration on TZM-bl cells as described above. The infectivity on primary rhesus CD4 T cells ranged between $1.83 \times 10^{-4}$ and $5.32 \times 10^{-5}$ IU/particle, which is similar to other TF SHIVs [51].

## Generation of plasma inoculum for in vivo competitions in RM 5695 and 5181

The inoculum stock for RM 5695 (1400 µl total) was generated from plasmas collected from seven timepoints from the SHIV.C.CH505-infected RM 6072, 6069, and 6070. 200 µl of plasma from RM 6072, from week 4 (VL:1,370,000 c/mL), week 10 (VL: 769,000 c/mL), and week 20 (VL: 69,000 c/mL); from RMs 6069 at week 10 (VL: 4,730,000 c/mL), and week 20 (VL: 3,710,000 c/mL), and from RM 6070, week 10 (VL: 5,740,000 c/mL), and week 20 (VL: 646,000 c/mL) were combined and inoculated intravenously. For passage 2, RM 5181 was intravenously inoculated with a total of 800 µl of plasma from RM 5695 at the following timepoints: 4, 8, 16, and 24 WPI (VLs of 1,480,000; 99,000; 16,000; 14,000 c/mL, respectively).

## 8 variant competition study inoculum

The eight-variant inoculum was prepared by adding 10 ng of each of the 8 candidate CH505 combination clones, with the stock containing 80 ng total per animal. The stock was then

diluted in complete media RPMI 1640 Media (Thermo Fisher Scientific #11875199) supplemented with 15% Fetal Bovine Serum (Thermo Fisher Scientific #A5256701), 100 U/ml penicillin-streptomycin (Thermo Fisher Scientific #15070063) to a volume of 1 mL per animal. RM T646, 5593, and 150250 were inoculated with the CH505adapted stock mixture.

### V2 vs TF competition inoculum

The V2 vs TF competition inoculum was prepared by combining 2 stocks: SHIV.C.CH505v2 and SHIV.C.CH505.TF. Two inocula were prepared, the first containing 15 ng of each variant (30 ng total), which was administered to RM R276 and T277. The second contained 150 ng of each variant (300 ng total) and was administered to RM T278 and T279. Inocula stocks were prepared using the same medium as above and sequenced by SGS as described above.

### Single genome sequencing

Single genome full-length gp160 env sequences were generated as previously described [51]. Briefly, 20,000 viral RNA (vRNA) copies were extracted from plasma virus using the Qiagen BioRobot EZ1 Workstation with EZ1 Virus Mini Kit v2.0 (Qiagen). Eluted vRNA was subsequently used as a template for cDNA synthesis and reverse-transcribed using the reverse primer SHIV.Env.R1 (5′- TACCCCTACCAAGTCATCA-3′) and SuperScript III reverse transcriptase (Thermo Fisher Scientific #18080044). cDNA was serially diluted to identify the dilution at which <30% of wells contained PCR amplicons of the correct size. The SHIV gp160 env genome was amplified via nested PCR with primers as follows: first round forward primer SHIV.Env.F1 (5′- CGAATGGCTAAACAGAACA-3′), second round forward primer SHIV. Env.F2 (CTACCAAGGGAGCTGATTTTC), first round reverse primer SHIV.Env.R1 (5′- TACCCCTACCAAGTCATCA-3′), and second round reverse primer SHIV.Env.R2 5′- TATTTTGTTTTCTGTATGCT-3'). PCR conditions were: for the first round of nested PCR: 94˚C, 2 min; 37x [94˚C, 20 sec; 55˚C, 30 sec; 68˚C, 3 min 30 sec]; 68˚C, 10 min and for the second round of nested PCR 94˚C, 2 min; 42x [94˚C, 20 sec; 54˚C, 30 sec; 68˚C, 3 min 30 sec]; 68˚C, 10 min. Amplicons were sequenced via the MiSeq platform (Illumina). Raw reads were aligned to the SHIV.C.CH505 reference using Geneious R9. Sequences that contained mixed bases at a frequency of >25% per nucleotide position were excluded from further analysis.

Sequences from plasma virus from RM 6069, 6070, and 6072 (Study 1) were generated at 4, 10 and 20 WPI. Sequences from RM in Study 2 were generated at 52 WPI. Due to the limited number of available sequences, which was attributable to low VL at the time of sampling in some RM, an 100% penetrance threshold was used to identify virus populations that had one or more non-synonymous AA substitutions in Env. Sequences were re-examined using 50% as the penetrance threshold for non-synonymous AA substitutions. Sequences from plasma virus from RM 5695 (Study 3) were generated at 2, 4, 12, 16, and 24 WPI. Sequences from plasma virus from RM 5181 (Study 3) were generated at 2, 8, 12, and 24 WPI. This manuscript includes previously published [39], that are presented with new sequences in Figs 1 and 3.

### Deep sequencing by illumina

Approximately $10^3$–$10^5$ viral RNA copies were extracted from plasma and cDNA was generated as described above. The volume of cDNA required to sample $10^{2-3}$ individual copies was determined and used as the input for bulk PCR amplification, with amount of plasma vRNA copies in the initial sample being the limiting factor. Nested bulk PCR conditions were used as follows for the first round of nested PCR: 94˚C, 2 min; 37x [94˚C, 20 sec; 50˚C, 30 sec; 68˚C, 1 min]; 68˚C, 10 min. For the second round of nested PCR, the PCR conditions were as follows: 94˚C, 2 min; 42x [94˚C, 20 sec; 68˚C, 30 sec; 68˚C, 1 min]; 68˚C, 10 min. The primers used

were as follows: first round forward primer CH505adapted F1 5'-TGCTCCAGCTGGT-TATGCG-3', second round forward primer CH505adapted F2 5'-TCGTCGGCAGCGTCA-GATGTGTATAAGAGACAGTGTCAGCACAGTACAATGTACACA-3', first round reverse primer CH505adapted R1 5'-TGTTATGTTTCCTGCAATGGG-3' and second round reverse primer CH505adapted R2 5′-GTCTCGTGGGCTCGGAGATGTGTATAAGAGACA-GATTGCTCGTCCCACCTCCTG-3′.

Bulk PCR reaction products were multiplexed and sequenced via the MiSeq platform (Ilumina). The frequency of variants most similar to the 8 candidate CH505adapted clones was determined using a software analysis pipeline, incorporating open-source software with custom scripts written in python, bash, and R. Raw reads were aligned to the SHIV.C.CH505 TF reference sequence using bowtie2 with default parameters. Reads that had more than 7 mismatches to the reference genome were excluded from downstream analysis, to allow for reads with all 6 mutations identified upstream (V255I, N279D, K302N, Y330H, N334S, H417R) plus one additional mismatch. Each remaining read's bowtie alignment was used to extract the nucleotides at the six variant locations. The 6 nucleotides of interest were used to classify each read, and each sample was condensed to a summary of how many reads had each a particular combination of nucleotides/variants. Frequencies of variants were calculated against the number of reads passing all filters.

## Rhesus CD4 T cell isolation and activation

Rhesus CD4 T cells were isolated as previously described [51]. Briefly, rhesus PBMC were combined with magnetic CD4 Microbeads (Miltenyi Biotec) and processed according to the manufacturer's instructions. Antibiotin MACSiBead particles loaded with biotinylated anti-CD2, CD28, and CD3 antibodies (T-cell activation/expansion kit from Miltenyi Biotec) were combined with isolated rhesus CD4 T cells as described by the manufacturer. The isolated CD4$^+$ cells were cultured in complete RPMI growth medium supplemented with 15% fetal bovine serum, 100 U/mL penicillin-streptomycin (Gibco), and 30 U/mL IL-2 for 5–6 days.

## SHIV replication in primary rhesus CD4 T Cells

SHIV infection of primary rhesus CD4 T cells was performed as described [51]. Each SHIV variant was normalized to 300 ng of input virus per well and combined with $2X10^6$ rhCD4 T cells in complete RPMI 1640 growth medium containing 40 μg/mL DEAE–dextran and 30 U/ml IL-2 to a total volume of 1 mL per well in a 24 well-plate. The cell and virus mixtures were incubated at 37˚C in 5% $CO_2$ for 4 h and subsequently washed 3 times with 10 mL of complete RPMI growth medium without DEAE-dextran and IL2. Cell and virus mixtures were each resuspended in 1 mL of complete RPMI1640 growth medium and 30 U/ml IL2 in 24-well plates and cultured for 9 days. Supernatants were collected on days 0, 3, 6, and 9 for p27 antigen measurement. For the summary graphs, the first timepoint at which the replication of TF. CH505 was >5 or 10 ng/ml for point and combination clone experiments, respectively, was used.

## TZM-bl neutralization assay

TZM-bl neutralization assays were performed as follows; TZM-bl cells, cultured in DMEM culture media as above described, were seeded at the density of 15,000 cells per well (96-well plate). After 24 hours, selected antibodies (Ab) were serially diluted 5-fold and mixed with 4,000 IU of virus stock. Antibody-virus mixtures were co-incubated for 1 h at 37˚C, 5% $CO_2$ and then added in triplicate to the pre-seeded TZM-bl cells. After 48 hours, cells were lysed in 0.5% Triton-DPBS for 1 hour, and then mixed with the luciferase substrate (Promega

Luciferase System #E1501) and analyzed on the Synergy Plate Reader system (BioTek). Background-corrected luciferase activity for each sample was determined and the IC50 values were calculated using the variable slope function in Prism (v7.0). Monoclonal antibodies were obtained from the NIH HIV Reagent program.

## ZB5 entry assay

ZB5 cells were propagated and infected as previously described [51]. Briefly, cells were propagated in complete growth medium (DMEM High Glucose Thermo Fisher Scientific #11965092) supplemented with Penicillin-Streptomycin-Glutamine (Thermo Fischer Scientific #10378016), 10% Fetal Bovine Serum (Thermo Fisher Scientific #A5256701), and either 5 μg/mL blasticidin (Thermo Fisher Scientific #A1113902) to select for rhCCR5 expression and 5 μg/mL puromycin (Sigma-Aldrich #P9620) to select for cells expressing rhCD4. The ZB5 entry assay was performed in triplicate. Cells were seeded at $1 \times 10^4$ per well in 100 μl of complete media in 96-well plates and incubated overnight. Virus stock dilutions were made to final concentrations in complete DMEM + 40 μg/mL DEAE–dextran to achieve 0.5 ng p27 antigen/well. On Day 1 medium was removed and cells were incubated at 37˚C, 5% $CO_2$ with the virus inoculum diluted in growth medium for 48h. After 48h, cells were lysed with Bright-Glo (Promega #E2620), and RLUs were measured on a Luminoskan Ascent luminometer (Thermo Fisher Scientific #5300330). Entry efficiency of each CH505 clone was determined via normalization to CH505.TF (set to 100%) following correction for background luciferase activity.

## Modeling the TF to V2 replicative capacity difference

The relative fitness of v2 compared to TF is assessed by calculating the difference in growth rate during the first 4 weeks of infection in animals from Study 5 Cohort. We fit a line via least squares to logit of the fraction viral load composed of the v2 variant, $m$, as evaluated via high throughput sequencing of the V255I site with a 1:1 dilution. That is, we fit the line $logit(m) = \ln(R_i) - (g_w - g_m)t$, where $R_i$ is the initial ratio of v2 to TF, $g_w - g_m$ is the difference in growth rate between TF and v2, and $t$ is time (in days). Here, we assume the difference in growth rate of TF and v2, $g_w - g_m$, is constant with time. The parameters $g_w - g_m$ and $R_i$ are estimated based on the sequencing for weeks 1, 2, and 4 and the sequencing of the corresponding inoculation (week 0) for each animal individually and across all animals.

## Intact Proviral DNA Assay (IPDA)

CD4+ cells were isolated from peripheral blood mononuclear cells (PBMC) and lymph nodes (LN) using the CD4+ T cell Isolation kit (Miltenyi Biotech #130-092-144). Genomic DNA (gDNA) was obtained by using the All-Prep DNA/RNA Mini kit (Qiagen # 80204). Digital droplet PCR was performed as previously described [26,71–73]. Briefly, 200–250 ng of DNA was added in each ddPCR reaction containing 11 μl of ddPCR 2x super mix (Bio Rad #1863024), 2 units of HindIII per reaction well (New England Biolabs # R0104S), 600 nM of each primer (Pol and ENV) and 200nM of each probe. ribonuclease P/MRP subunit p30 (RPP30) was used to determine the DNA Shearing Index. Primer sequences (indicated as 5' to 3') were Pol Fw GCAGGGATAGAGCACACCTTTG, Pol Rev CTATGGTTTCTACT-GAATTTGCTTGTTC, Pol Intact probe FAM-TTTCAGGTGGTGATTCA-MGB-NFQ, Pol hypermutated probe TAGGTGGTGATTTATT-MGB-NFQ, Env Fw AGTGGTGGAGAGA-GAAAAAAGAGC, Env Rev GTCTGGCCTGTACCGTCAGC, Env probe HEX-CCTTGGGTTCTTGGGA-MGB-NFQ, Two RPP30 amplicons have been used to determine the DNA Shearing Index (DSI) with an aliquot of each DNA sample (5 ng) added to the

same master mix mentioned above with RPP30 specific primers (500 nM) and probes (250 nM). Primers were RPP30 Amplicon 1 Fw AGGATGCTCCGGGAGTATGTA, RPP30 Amplicon 1 Rev CCTGCTTGTCACCTATATAACAT, RPP30 Amplicon 2 Fw ACAGACTCACACAATTTAGG, RPP30 Amplicon 2 Rev ACATTCATGCCACTGCACTC, RPP30 Amplicon 1 Probe FAM-TCAAGCTGGGAGACGGAAGAGTCAGT-MGB-NFQ, Amplicon 2 Probe HEX-ACAGGGTCTCACTTTGTTGTCCA-MGB-NFQ. The Output of intact and total SHIV copy numbers were normalized to copies per million cells.

## Molecular modeling of HIV-1 CH505 envelope

The coordinates of the X-ray structure of M1214_N1 Fab in complex with CH505 TF chimeric SOSIP.664 Env trimer (PDB ID 6VY2) were used to build an all-atom fully glycosylated trimer model. Using the CHARMM-gui web server [74–77], missing envelope regions were modeled and all the surface exposed N-linked glycosylation sites (PNGS) were modeled to have a mannose-5 sugar moiety at each N-linked glycosylation sequon along the protein sequence [78]. 1gc1 (resolution 2.5 Å) was used to determine the CD4-bs residues. All gp120 residues with a buried surface area $> 5$ Å$^2$ in the interface with CD4 in the structure were colored orange on the CH505 env structure (6vy2). 6meo (resolution 3.9 Å) was used to determine the CCR5-bs residues. All gp120 residues with a buried surface area $> 5$ Å$^2$ in the interface with CCR5 in the structure were colored yellow on the CH505 env structure (6vy2). For Fig 4B, residue positions were shown as spheres and color coded in PyMol [79] according to their Shannon entropy across a 208 strain panel of HIV-1 envelopes. This panel was made of geographically and genetically diverse strains representing the major subtypes and circulating recombinant forms of HIV-1 [80]. Shannon entropy was calculated with a custom python script. Conformational entropy was color-coded based on residual entropy and mutations were represented as spheres.

## Env sequences

The full list of sequences used within the current study is provided in the supplemental data (S2 File).

## Supporting information

**S1 Fig.** (**A**) Viral kinetics in three RMs infected with SHIV.C.CH505 with lower viral loads. The dotted line indicates the assay's limit of detection (LoD) (62 copies/ml). (**B**) Frequency of signature mutations determined by SGS.
(TIF)

**S2 Fig. Variant frequencies determined by gp160 *env* SGS at weeks 1 and 2 post-infection in three RM inoculated with eight variants.** All eight variants were identified, with the 5MA variant replicated most efficiently as observed by Illumina deep sequencing (Fig 7). No recombinant viruses were identified. The distribution of variants in the inoculum (left side) was determined by deep sequencing and reproduced from Fig 7. S2 Fig was created with BioRender.com.
(TIF)

**S3 Fig. Mutational frequency observed in single genome sequencing (SGS) of gp160 env on longitudinal plasma timepoints.** The frequency of each of the 5 signature mutations, as well as N130D and N279D, is shown in the heatmap, with the number of SGS sequences shown at right. S3 Fig was created with BioRender.com.
(TIF)

**S4 Fig. Neutralization sensitivity to prototypic bnAbs as determined by TZM.bl assay for SHIV.C.CH505 (CH505.TF), SHIV.C.CH505.v2 (CH505.v2), and MLV (negative control).** The concentration (μg/mL) of bnAb conferring a 50% reduction in infectivity (IC50) is shown in the table for each bnAb.
(TIF)

**S1 File. Complete list of mutations observed for RM 5695 by week 24.**
(XLSX)

**S2 File. List of sequences analyzed within the study.**
(TXT)

## Author Contributions

**Conceptualization:** Anya Bauer, Ivelin Georgiev, Barbara Felber, Brandon F. Keele, Ronald Veazey, George M. Shaw, Katharine J. Bar.

**Data curation:** Anya Bauer, Francesco Elia Marino, Jaimy Joy, Kevin McCormick, Katharine J. Bar.

**Formal analysis:** Steffen S. Docken, Miles P. Davenport, Katharine J. Bar.

**Funding acquisition:** Katharine J. Bar.

**Investigation:** Anya Bauer, Emily Lindemuth, Hui Li.

**Methodology:** Anya Bauer, Emily Lindemuth, Francesco Elia Marino, Ryan Krause, Steffen S. Docken, Suvadip Mallick, Kevin McCormick, Clinton Holt, Ivelin Georgiev, Brandon F. Keele, Miles P. Davenport, Hui Li, Katharine J. Bar.

**Project administration:** Francesco Elia Marino, Katharine J. Bar.

**Resources:** Francesco Elia Marino, Barbara Felber, Katharine J. Bar.

**Software:** Francesco Elia Marino.

**Supervision:** Katharine J. Bar.

**Validation:** Katharine J. Bar.

**Visualization:** Katharine J. Bar.

**Writing – original draft:** Anya Bauer, Francesco Elia Marino, Katharine J. Bar.

**Writing – review & editing:** Francesco Elia Marino, Jaimy Joy, Steffen S. Docken, Suvadip Mallick, Kevin McCormick, Ivelin Georgiev, Barbara Felber, Brandon F. Keele, Ronald Veazey, Miles P. Davenport, Hui Li, George M. Shaw, Katharine J. Bar.

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
