## [Decision Letter · Decision Letter 0]

23 Jan 2023

Dear Dr Bar,

Thank you very much for submitting your manuscript "Adaptation of a transmitted/founder simian-human immunodeficiency virus for enhanced replication in rhesus macaques" for consideration at PLOS Pathogens. As with all papers reviewed by the journal, your manuscript was reviewed by members of the editorial board and by several independent reviewers. In light of the reviews (below this email), we would like to invite the resubmission of a significantly-revised version that takes into account the reviewers' comments.

We cannot make any decision about publication until we have seen the revised manuscript and your response to the reviewers' comments. Your revised manuscript is also likely to be sent to reviewers for further evaluation.

Sincerely,

Daniel C. Douek

Academic Editor

PLOS Pathogens

Richard Koup

Section Editor

PLOS Pathogens

Kasturi Haldar

Editor-in-Chief

PLOS Pathogens

orcid.org/0000-0001-5065-158X

Michael Malim

Editor-in-Chief

PLOS Pathogens

orcid.org/0000-0002-7699-2064

Reviewer's Responses to Questions

**Part I - Summary**

Reviewer #1: In this very well-written manuscript, Bauer and colleagues thoroughly outline the rationale for and the development and in vitro / in vivo validation of a novel T/F SHIV for use in nonhuman primate cure studies with interventions that target the HIV envelope. Using plasma from macaques infected with SHIV.C.CH505 the authors apply single genome sequencing to find mutations associated with higher set point viral loads (as low viral set points and spontaneous control of viremia has been a criticism of this and other SHIV strains). The impact of these mutations in terms of replication and antibody sensitivity is then tested individually and in combination using infectious molecular clones. The authors identify 5 mutations within or near the CD4 and coreceptor binding sites that distinguish the original SHIV.C.CH505 from an adapted SHIV.C.CH505.v2 and demonstrate that this mutated virus displays enhanced virus entry and replication. Importantly, spontaneous control of viral replication (in absence of ART) was not found. After 6 months of ART, SHIV.C.CH505.v2 rebounded soon after ART interruption and set point viral loads after rebound were similar to or ~1 log lower than pre-ART set point.

Reviewer #2: In this report, Bauer et al., have illustrated the development of a minimally adapted simian-human immunodeficiency virus, named SHIV.C.CH505.v2. They have conducted detailed sequence analyses to first identify a signature of envelope mutations associated with higher viremia in infected rhesus macaques. And used short-term in vivo mutation selection and competition to identify an isolate with five amino acid changes that improved in vivo virus replication fitness in macaques. They show that this minimally adapted SHIV had improved virus entry, enhanced replication, and preserved neutralization profiles in vitro. Also, their results demonstrate the virus rapidly outcompeted the parental SHIV and most importantly, persisted during suppressive antiretroviral therapy by showing rebound at treatment interruption.

Overall, the manuscript is very well written, clearly organized and the results are important in the development of an enhanced tool for NHP studies of HIV-1 transmission, pathogenesis, and cure. My minor comments are listed below:

1. Lines 84-86: “However, differences between SIVmac and HIV-1, including highly divergent envelope (env) sequences, distinct accessory genes, and variable set-point viremia, immunopathogenesis and natural history, challenge the direct translation of NHP study findings to people living with HIV-1 (PLWH).” This statement conveys that the immunopathogenesis of SIVmac and HIV-1 are distinct, which can be misleading to the general reader as the SIVmac model is a widely used and well-accepted model of HIV immunopathogenesis. I would suggest removing “immunopathogenesis” from the multiple other factors listed here as being divergent between SIVmac and HIV-1.

2. Line 180: Figure.2 legend mentions that a setpoint of > 10^4 copies/ml was used as the criterion for defining high VL, but the VL fluctuate a lot and some high VL seem to be below that threshold for many time-points. Was a particular time-point used? Please clarify.

3. Lines 318-319: “All animals became productively infected with peak VL ranging from

10^4 to 10^6 c/ml (Fig. 7B), though it is probable that the peak VL was not captured with this sampling strategy”. What time-point was this? If it is earlier than 2 weeks, then the authors may just state the time-point and add that the peak viremia was likely higher than this.

Reviewer #3: There is great interest in the use of bNAbs for HIV therapeutics and cure, however, many of the available SHIVs do not have replication kinetics similar to that of HIV. Bauer et al includes data from previous studies to identify a panel of mutations in env that were repeatedly observed in different animals and suspected to improve replication. They then go on to evaluate these mutations in vitro to assess their replication and neutralization properties. They then evaluated combinations of these mutations both in vitro and in vivo. Finally, in order to assess whether the new model is an improvement, they competed the new and parental model in vivo in four animals. Unfortunately, the in vivo replication of these clones was only evaluated over the first four weeks of infection. As a result, it is impossible to assess whether the new model is an improvement over extended periods of untreated infection. The authors have identified an intriguing new model that warrants additional investigation and may prove to be useful for future studies.

**Part II – Major Issues: Key Experiments Required for Acceptance**

Reviewer #1: The major criticism of this work is the small number of macaques used for the final experiment of SHIV.C.CH505.v2 persistence on ART and rebound after ART interruption. They start with 4 macaques total, but only 2 were treated with ART for a meaningful period of time and then allowed to rebound. This limitation must be acknowledged. Additionally, suppression of viremia within 2 weeks of ART initiation is unusual for both SIV and HIV infections, even with integrase-containing regimens. This finding should be considered in the Discussion of this model.

Reviewer #2: None.

Reviewer #3: • The authors work very hard to characterize the signature mutations that they identified, but then spend much less time comparing SHIV.C.CH505.V2 to SHIV.C.CH505 TF. There is extensive information about the infection dynamics of SHIV.C.CH505 TF. As a result, one of the most important ways to evaluate the new model is to compete it against SHIV.C.CH505 TF. The authors do this in four animals, but only assess the relative frequency of each clone up to week 4. At this time the population is dominated by SHIV.C.CH505.V2 but after the host immune response arises the relative fitness of the clones could change. In addition, the high viral loads in figure 8B cannot just be attributed to SHIV.C.CH505.V2 alone. The manuscript would be much stronger with thorough analysis of the viral population after a longer period of time. Given that recombination and evolution are likely to have fundamentally changed the population, this would require SGS sequencing and careful analysis of the population. In the absence of this data the following statement doesn't seem accurate/precise "In vivo, the minimally adapted virus rapidly outcompetes the parental SHIV". A more accurate description would be that "In vivo, the minimally adapted virus outcompeted the parental SHIV over the first few weeks of infection, but their relative fitness was not evaluated over extended periods of untreated infection". Similarly, it would be very helpful to know the relative frequencies of the different combinations of clones (Figure 7) after a longer period of time.

• I think that multiple names are being used for the same stocks (SHIV.C.CH505, SHIV.C.CH505.TF, TF SHIV.C.CH505, TF SHIV, CH505-TF, parental SHIV.C.CH50, TF). If these are the same, then it would be easier if a single name was used. If they aren't the same then the text needs to be clarified.

• Was a UMI used in the MiSeq analyses? If not, how can the frequency of an allele be assessed without knowing the number of RNA templates sequenced? This is particularly problematic when populations contain many clonal sequences making it impossible to determine whether the population contains many identical genomes or the same small number of genomes are being sequenced many times. In the absence of a UMI, the frequency of genomes cannot be accurately assessed.

• It was challenging to identify whether the data were new or previously published. Specifically, was the data in figures 1, 2 and 3 previously published? If so, it may streamline the study to say that the 7 signature mutations repeatedly emerged during previous infection experiments.

• The source of the clones should be clarified. When describing Figure 5 the authors state " We next generated IMCs encoding combinations of the six signature mutations in a newer TF SHIV backbone, 3C, which contains deletions in the tat and gp41 env reading frames that were selected in vivo in multiple RM infected with distinct TF SHIVs and conferred a fitness advantage in vitro compared to the previous backbone (63), and has been incorporated into current iterations of TF SHIVs (54)." That suggests that all of the signature mutations were identified in experiments infecting RM with the original SHIV.C.CH505 IMC and its descendants that were subsequently passaged in RM. While the mutations accumulated in the older backbone I think that they were often (always?) characterized using the newer 3C backbone. This should all be discussed more clearly. In addition, it seems possible that the mutations may behave differently in the two backbones. It is definitely worth discussing the implications of this change and specifying the source of the variants used in the different experiments.

**Part III – Minor Issues: Editorial and Data Presentation Modifications**

Reviewer #1: 1. Page 6, line 152 appears to be missing a reference in which SHIV.C.CH505 was used in adult macaques (Dashti et al J Virol 2020).

2. Page 6, line 152: correct SHIC to SHIV.

3. Page 7: text referring to Fig 2B defines fully penetrant as 100%; however, the 3 mutations that are stated to meet this criterion (K302N, Y330H, H417R) are shown at a frequency of 80%. The text should be revised to avoid confusion.

4. Page 7, line 181: correct 80% to 83%.

5. Fig 1C legend (and other legends describing similar figures): specify that percentages are calculated from the total number of sequences generated (to distinguish from the percentage of animals with the specific mutation detected).

6. Highlighter plots: what do the black ticks represent?

7. Fig 3: the rationale for performing SGS out to week 48 in RM 5695 and week 24 in RM 5181 (and not from later time points) should be clearly stated.

8. Fig 3B: there appear to be additional mutations that arise beyond the 7 shown in Fig 3C. Page 8, Lines 202-203 states that no other mutations rose to full penetrance by week 24 although the data are not presented to evaluate this statement. Perhaps a table could be included with all of the mutations and their penetrance at week 24?

9. Results text describing Fig 6 should detail the panels being described (A-F) to enhance clarity.

10. Fig 6C-D: CH505.v2 is included here but has not yet been described in the text. Suggest removal.

11. Page 13, line 371: specify if the individual or combination variants are used for infection.

12. Page 13, line 319: explain why the peak viral load may not have been captured with the sampling strategy (that occurred “frequently over the first 4 weeks of infection”) or remove this phrase.

13. Fig 7A and 8A, shouldn’t the doughnut shapes representing the inoculum show an equal proportion of each variant?

14. Was there any difference in peak viremia for v2 with and without barcoding? The x-axis scale of Fig 8B make this difficult to assess.

15. Results text describing figure 9A should specify the exact number of weeks until rebound (and the definition of rebound used) after ART interruption.

16. Limit of detection of plasma viral load assay used should be added to the Methods and shown on the figures.

17. I did not see legends for the supplementary figures but may have missed them.

Reviewer #2: 1. Lines 84-86: “However, differences between SIVmac and HIV-1, including highly divergent envelope (env) sequences, distinct accessory genes, and variable set-point viremia, immunopathogenesis and natural history, challenge the direct translation of NHP study findings to people living with HIV-1 (PLWH).” This statement conveys that the immunopathogenesis of SIVmac and HIV-1 are distinct, which can be misleading to the general reader as the SIVmac model is a widely used and well-accepted model of HIV immunopathogenesis. I would suggest removing “immunopathogenesis” from the multiple other factors listed here as being divergent between SIVmac and HIV-1.

2. Line 180: Figure.2 legend mentions that a setpoint of > 10^4 copies/ml was used as the criterion for defining high VL, but the VL fluctuate a lot and some high VL seem to be below that threshold for many time-points. Was a particular time-point used? Please clarify.

3. Lines 318-319: “All animals became productively infected with peak VL ranging from

10^4 to 10^6 c/ml (Fig. 7B), though it is probable that the peak VL was not captured with this sampling strategy”. What time-point was this? If it is earlier than 2 weeks, then the authors may just state the time-point and add that the peak viremia was likely higher than this.

Reviewer #3: • There are two sequencing approaches described in this manuscript (SGS and MiSeq), but it's not always clear which is being used. I think that the frequency data in Figs 1C, 2B, 3C is based on SGS and 7A, 8A and 8C are all MiSeq. I would specify this in the legends. Also, I would list the limit of detection for the different assays.

• The authors are describing a new SHIV with many properties that may make it desirable for future studies. The study culminates with an experiment infecting four RM with the newly developed SHIV.C.CH505.v2. At the very end of the results they mention that 2 of the 4 animals had to be euthanized due to wasting after approximately 5 years of therapy. In addition, RM 5695 and RM 5181 had to be euthanized between 1 and 2 years after infection. I would add symbols to the viral load graphs indicating when the animals were euthanized. This could be helpful information as researchers are evaluating whether this new model is appropriate for the study that they are planning.

• Please list the specific env that is serving as the master sequence in the highlighter plots in Fig 1A and 3B. SHIV.C.CH505 seems the best choice to me.

• The authors should provide justification for the various assays. Why were some done with RM CD4+ T cells, others with ZB5 cells and others with TZMs?

• It would be interesting to discuss why the entry patterns in Fig 5C and 6B are so different.

• The authors state that the signature mutations "likely arose via escape from early immune responses or reversion to group M consensus". What evidence suggests that 255, 279 and 417 arose this way?

• The discussion of the signature mutations (lines 225-235) is confusing. The authors state "Three signature mutations (K302N, Y330H, N334S) encode amino acids that are highly conserved (67-99%) (60)" but go on to state "but the signature mutation residues are less commonly seen than the TF SHIV.C.CH505 amino acid (255 and 279), or neither mutation is common (417)."

• Is sensitivity to two bNAbs sufficient to assess the neutralization tier? My understanding was that the standard method also involved assessing sensitivity to polyclonal IG (PMID: 19939925)

PLOS authors have the option to publish the peer review history of their article (what does this mean?). If published, this will include your full peer review and any attached files.

Reviewer #1: No

Reviewer #2: **Yes: **Namita Rout

Reviewer #3: No
---

## [Decision Letter · Decision Letter 1]

11 Jun 2023

Dear Dr Bar,

Thank you very much for submitting your manuscript "Adaptation of a transmitted/founder simian-human immunodeficiency virus for enhanced replication in rhesus macaques" for consideration at PLOS Pathogens. As with all papers reviewed by the journal, your manuscript was reviewed by members of the editorial board and by several independent reviewers. The reviewers appreciated the attention to an important topic. Based on the reviews, we are likely to accept this manuscript for publication, providing that you modify the manuscript according to the review recommendations.

Reviewer 3 has a few minor concerns and suggestions which you should easily be able to address in the text  

Sincerely,

Daniel C. Douek

Academic Editor

PLOS Pathogens

Richard Koup

Section Editor

PLOS Pathogens

Kasturi Haldar

Editor-in-Chief

PLOS Pathogens

orcid.org/0000-0001-5065-158X

Michael Malim

Editor-in-Chief

PLOS Pathogens

orcid.org/0000-0002-7699-2064

Reviewer Comments (if any, and for reference):

Reviewer's Responses to Questions

**Part I - Summary**

Reviewer #1: The authors have satisfactorily responded to all of my prior questions and suggestions.

Reviewer #2: This manuscript describes the development of a minimally adapted simian-human immunodeficiency virus, named SHIV.C.CH505.v2. The results show that this minimally adapted SHIV had improved virus entry, enhanced replication, and preserved neutralization profiles in vitro. Also, they demonstrate the virus outcompeting the parental SHIV and persistence during suppressive antiretroviral therapy .

The manuscript is very well written, clearly organized and the results are clearly important in the development of an enhanced tool for NHP studies of HIV-1 transmission, pathogenesis, and cure.

Reviewer #3: I greatly appreciate the efforts that the authors made to address the reviewers' comments. The revised manuscript is much improved and will make a substantial contribution to the field.

**Part II – Major Issues: Key Experiments Required for Acceptance**

Reviewer #1: none

Reviewer #2: None

Reviewer #3: (No Response)

**Part III – Minor Issues: Editorial and Data Presentation Modifications**

Reviewer #1: none

Reviewer #2: All addressed by the authors.

Reviewer #3: 1. I understand the authors justification for presenting frequency data from deep sequencing analyses that lack UMIs. It would be helpful to describe those justifications in the limitations section in the discussions. This would help the average reader understand the rationale behind their experimental design.

2. Panels in Fig 5 are discussed out of order

3. The following sentence should be clarified "Thus, we used the SGS to screen for evidence of in vivo recombination. SGS env sequences from weeks 1 and 2 (n=162) identified all eight inoculum variants, but no in vivo recombinants (Fig S2); thus, we excluded recombinant sequences from the Illumina sequencing analyses."

PLOS authors have the option to publish the peer review history of their article (what does this mean?). If published, this will include your full peer review and any attached files.

Reviewer #1: No

Reviewer #2: **Yes: **Namita Rout

Reviewer #3: No

Figure Files:

Data Requirements:

Reproducibility:

References:

---

## [Editor Report · Decision Letter 2]

21 Jun 2023

Dear Dr Bar,

We are pleased to inform you that your manuscript 'Adaptation of a transmitted/founder simian-human immunodeficiency virus for enhanced replication in rhesus macaques' has been provisionally accepted for publication in PLOS Pathogens.

Best regards,

Daniel C. Douek

Academic Editor

PLOS Pathogens

Richard Koup

Section Editor

PLOS Pathogens

Kasturi Haldar

Editor-in-Chief

PLOS Pathogens

orcid.org/0000-0001-5065-158X

Michael Malim

Editor-in-Chief

PLOS Pathogens

orcid.org/0000-0002-7699-2064
---

## [Editor Report · Acceptance letter]

28 Jun 2023

Dear Dr Bar,

We are delighted to inform you that your manuscript, "Adaptation of a transmitted/founder simian-human immunodeficiency virus for enhanced replication in rhesus macaques," has been formally accepted for publication in PLOS Pathogens.

Best regards,

Kasturi Haldar

Editor-in-Chief

PLOS Pathogens

orcid.org/0000-0001-5065-158X

Michael Malim

Editor-in-Chief

PLOS Pathogens

orcid.org/0000-0002-7699-2064